# EEVR: A Dataset of Paired Physiological Signals and Textual Descriptions for Joint Emotion Representation Learning

**Pragya Singh**
IIIT-D, New Delhi, India
pragyas@iiitd.ac.in

**Ritvik Budhiraja**
*IIIT-D, New Delhi, India
ritvik19322@iiitd.ac.in

**Ankush Gupta**
*IIIT-D, New Delhi, India
ankush21232@iiitd.ac.in

**Anshul Goswami**
IIIT-D, New Delhi, India
anshul20361@iiitd.ac.in

**Mohan Kumar**
RIT, Rochester, New York, US
mjkvcs@rit.edu

**Pushpendra Singh**
IIIT-D, New Delhi, India
psingh@iiitd.ac.in

## Abstract

*EEVR* (Emotion Elicitation in Virtual Reality) is a novel dataset specifically designed for language supervision-based pre-training of emotion recognition tasks, such as valence and arousal classification. It features high-quality physiological signals, including electrodermal activity (EDA) and photoplethysmography (PPG), acquired through emotion elicitation via 360-degree virtual reality (VR) videos. Additionally, it includes subject-wise textual descriptions of emotions experienced during each stimulus gathered from qualitative interviews. The dataset consists of recordings from 37 participants and is the first dataset to pair raw text with physiological signals, providing additional contextual information that objective labels cannot offer. To leverage this dataset, we introduced the Contrastive Language Signal Pre-training (CLSP) method, which jointly learns representations using pairs of physiological signals and textual descriptions. Our results show that integrating self-reported textual descriptions with physiological signals significantly improves performance on emotion recognition tasks, such as arousal and valence classification. Moreover, our pre-trained CLSP model demonstrates strong zero-shot transferability to existing datasets, outperforming supervised baseline models, suggesting that the representations learned by our method are more contextualized and generalized. The dataset also includes baseline models for arousal, valence, and emotion classification, as well as code for data cleaning and feature extraction. Further details and access to the dataset are available at https://melangelabiiitd.github.io/EEVR/.

## 1 Introduction

Recently, there has been a growing emphasis on maintaining mental well-being, as good mental health is essential for daily functioning and overall quality of life (Izutsu et al. (2015)). However, continuously monitoring mental health can be challenging, particularly with the demands of busy schedules. This lack of consistent monitoring can lead to the development of serious mental health

---

*Both the authors have equal contributions.

38th Conference on Neural Information Processing Systems (NeurIPS 2024) Track on Datasets and Benchmarks.

issues, which can significantly impact one's life. Various wearable devices and mobile applications have been developed in the past to help monitor mental well-being. In these applications, various kinds of physical and behavioural data are collected to serve as proxies for assessing shifts in mental well-being. These include facial images, facial videos, audio recordings, text, mobile phone data, and physiological signals (Wang et al. (2014); Thieme et al. (2020)). Lately, Deep Learning (DL) and Machine Learning (ML) techniques have been increasingly employed to predict various aspects of mental well-being using these proxies (Thieme et al. (2020); Saganowski et al. (2023); Ghandeharioun et al. (2017)). While most of the present proxies have been employed in the past to train DL and ML models, creating a robust model for everyday use necessitates modalities that don't impede users' movement or privacy and seamlessly integrate with their lifestyles. Physiological signal-based data offers these advantages over other modalities while mitigating potential data manipulation by subjects, a concern present with other modalities.

Present methods of collecting physiological emotion data typically rely on self-reports using standard emotion questionnaires or scales based on emotional theories or stimulus-based labels (e.g., data collected during a relaxing video is annotated as no stress). Previous studies include objective scales such as Visual Analogue Scales (VAS), the Positive and Negative Affect Schedule (PANAS), the Self-Assessment Manikin (SAM) scale, the Likert scale for basic emotions, and standardized questionnaires like the State-Trait Anxiety Inventory (STAI) are commonly used in previous studies. However, stimulus-based labels, which annotate data based on the type of stressor, often fail to reflect the subject's true emotional state. Similarly, objective annotations alone do not capture the nuanced details of emotions and are prone to human error. These methods may miss key emotional experiences, such as mixed or fleeting emotions, and often fail to capture the absence of emotion altogether, limiting the accuracy and depth of emotion recognition.

To address these limitations, this study introduces *EEVR*, a physiological signal-based emotion dataset collected in laboratory settings using 360° VR audiovisual stimuli. EEVR includes data from the two most commonly available physiological sensors in commercial wearable devices—Photoplethysmography (PPG) and Electrodermal Activity (EDA)—which have been widely collected in previous datasets. Emotional annotations were obtained through subjective evaluations using the PANAS and SAM emotion scales, along with self-reported raw textual descriptions of emotions felt by subjects during stimulus exposure. These descriptions were gathered through semi-structured qualitative interviews, providing a more contextualized understanding of emotions and allowing participants to elaborate on their emotional experiences in detail. EEVR is the first dataset to collect raw textual data for broader supervision, capturing the presence or absence of emotions experienced during the stimulus. This approach to collecting subjective textual responses to emotions has not been explored before. EEVR includes data from 37 participants who experienced emotions across all four quadrants of Russell's circumplex model. Additionally, it contains personality scores for each subject, collected using the Big Five Inventory 10 Item Scale (BFI-10) (Rammstedt et al. (2013)), and the psychological well-being details of each subject using the General Health Questionnaire-12 (GHQ-12) (Gureje and Obikoya (1990)).

Through this work, we make the following contributions:

- A novel multimodal physiological signal dataset collected in an immersive lab setting with aligned raw textual descriptions of emotions felt and self-reported valence and arousal scores.
- A readily replicable experimental procedure for capturing physiological response and textual descriptions within lab settings.
- We provide guidance on utilizing the dataset, along with open-source access to baseline models and the Contrastive Language-Signal Pre-training (CLSP) models, which leverage text supervision to learn more contextualized representations of emotions by combining physiological signals with text data.

## 2    Related Work

We have compared our dataset with previous datasets that have collected physiological signals for emotion recognition, as shown in Table 1. Previous works have combined physiological data with other modalities like video and audio. For instance, the MAHNOB-HCI (Soleymani et al. (2012)) dataset includes facial videos, audio inputs, eye-gaze data, and physiological signals from both the

| Dataset | #Subjects | Stimuli | Data Modalities | Annotations |
|---|---|---|---|---|
| MANHOB HCI | 27 | Audiovisual | ECG, GSR/EDA, RESP, TEMP, EYE GAZING, EEG, Facial Expressions and Audio. | Emotions, Arousal, Valence, Dominance. |
| DEAP | 32 | Music Video clip | EEG, ECG, PPG, GSR/EDA, EMG (Trapezius, Zygomaticus Muscle). | Arousal, Valence, Liking, Dominance and Familiarity. |
| WESAD | 15 | TSST, Audiovisual | ECG, EDA, EMG, BVP, Respiration, Temperature, Acceleration. | Stressor-based, PANAS, STAI, SAM. |
| CLAS | 62 | Cognitive load, Audiovisual | ECG, PPG, EDA, Acceleration. | SAM. |
| SWELL-KW | 25 | Office work with interruptions and time pressure | ECG, EDA, Face and upper body video, Posture, Computer logging. | NASA task load, SAM, Stress. |
| EMOGNITION | 43 | Audiovisual | PPG, EDA, BVP, Temperature, Acceleration, cardiac output measurement, Facial Expression. | Arousal, Valence, Avoidance Approach Motivation, Emotions. |
| AMIGOS | 40 | Audiovisual | EEG, ECG, EDA/GSR. | PANAS, SAM, Liking, Familiarity, Personality, Emotions. |
| ASCERTAIN | 58 | Audiovisual | GSR, Frontal EEG, ECG, Facial Landmarks. | SAM, Familiarity, Personality, Emotions. |
| DREAMER | 23 | Audiovisual | EEG, ECG. | SAM. |
| KEMOCON | 32 | 10 Minute long paired debate on social issues. | PPG, EDA, BVP, Temperature, Acceleration, EEG, ECG. | Arousal, Valence, Emotional Labels. |
| BIRAFFE2 | 103 | Music, Images, Games | ECG, EDA, Gamepad Acceleration, Gyroscope. | SAM, Personality, Game experience. |
| CASE | 30 | Audiovisual | ECG, BVP, EMG, EDA/GSR, Respiration, Temperature. | SAM. |
| StressID | 65 | Cognitive load Audio-Visual Public Speaking | ECG, EDA, Respiration, Speech, face video. | SAM, Stress. |
| VREED | 34 | 360 degree VR | ECG, EDA, Eye Tracking. | SAM, Emotions. |
| **EEVR (Ours)** | **37** | **360 degree VR** | **PPG, EDA.** | **Emotions, Arousal, Valence, Dominance Familiarity, Liking, Personality, GHQ-12, Textual Description.** |

Table 1: EEVR in comparison with other related datasets

peripheral and central nervous systems. Similarly, the DEAP (Koelstra et al. (2012)) dataset was collected using musical video clips in laboratory settings containing physiological signal data. However, the controlled environment of these datasets limits their ecological validity. The EMOGNITION (Saganowski et al. (2022)) dataset focuses on nine discrete emotions and collects both dimensional and discrete emotion annotations, emphasizing positive emotions, which are often overlooked. Other notable datasets that gather physiological signals include AMIGOS (Miranda-Correa et al. (2021)), ASCERTAIN (Subramanian et al. (2018)), DREAMER (Katsigiannis and Ramzan (2017)), and CASE (Sharma et al. (2019)). These datasets typically use non-immersive audiovisual stimuli in lab settings to elicit emotions, which may not reflect real-world experiences. Additionally, datasets like WESAD (Schmidt et al. (2018)), CLAS (Markova et al. (2019)), KEMOCON (Park et al. (2020)), StressID (Chaptoukaev et al. (2024)) SWELL-Knowledge Work (Koldijk et al. (2014)), and BIRAFFE2 (Kutt et al. (2022)) collect peripheral physiological data through constrained tasks alongside audiovisual stimuli. These tasks include the Trier Social Stress Test (TSST), logic and math problems, the Stroop test, debates on social topics, public speaking, office work with interruptions, and playing games. Although these tasks can induce stress, they often restrict emotional responses to specific settings and fail to capture a broad range of emotions. Further constrained settings in real life are also used for collecting emotion data, such as the NURSE dataset collected in hospital settings during COVID 19 (Hosseini et al. (2022)). VR-based emotion elicitation has recently become popular due to its higher ecological validity and capability to provide high immersion. For example, the VREED dataset (Tabbaa et al. (2022)) includes physiological signals and eye-gaze data from 34 participants collected

using VR-based audiovisual stimuli, providing higher immersion than video or audio-based stimuli in lab settings. Further existing literature has also highlighted the role of perceptions in emotion data (Markowitz and Bailenson (2023); Diemer et al. (2015); Barrett et al. (2011)), as how a person interprets or perceives a stimulus or real-life situation directly influences their emotional response to it, which is often not captured in objective labels. Moreover, recent theories of emotion, such as the appraisal theory and constructivist theory of emotions, have also emphasized the importance of cognitive processes in shaping emotional experiences Lazarus (1991); Barrett and Russell (2014) that is often not captured in prior dataset collection. Perception in prior work is often associated with collecting context data in the form of personality details Subramanian et al. (2018), activity data Gjoreski et al. (2017) or context as scene Thuseethan et al. (2022). However, none of the prior work has collected or considered participant's perceptions as context. This suggests the importance of collecting elaborate textual descriptions capturing participants' perceptions towards the stimulus as part of emotion data collection. EEVR aims to fill this gap by providing a novel dataset that aligns qualitative textual descriptions of participants' perspectives for each recorded physiological signal segment.

**Motivation for Paired Textual Descriptions.** Supervision through language or text has become a focal point in computer vision after the introduction of CLIP (Radford et al. (2021)). The emergence of large language models has also spurred an increase in research exploring language-guided supervision. This trend extends beyond vision and language, with modalities like audio (Elizalde et al. (2022)) and video (Wang et al. (2023)]) leveraging language supervision for pre-training to enhance generalization and usability. Concurrently, text-based pre-training methods have revolutionized the NLP domain in recent years. Despite the prominence of such approaches in various fields, there remains a notable gap in utilizing language-guided supervision for emotional recognition through physiological signals. While previous research has looked into leveraging text data (such as social media posts, text messages, and suicide notes) for emotion recognition (Thieme et al. (2020)), none of these efforts involved recording self-reported emotional descriptions from subjects alongside the collection of physiological signal data. Therefore, the EEVR dataset is an initiative in this regard, prompting new avenues for collecting emotional data based on physiological signals.

## 3 EEVR Dataset

### 3.1 Experimental Protocol

Our experiment protocol to collect the EEVR dataset is illustrated in Figure 1. The experiment was conducted using VR 360° audiovisual stimuli. The stimuli consist of *N=8* short videos (two videos from each quadrant of the Russell circumplex model (Russell et al. (1989))) covering all emotions. Next, we explain the data collection procedure as follows:

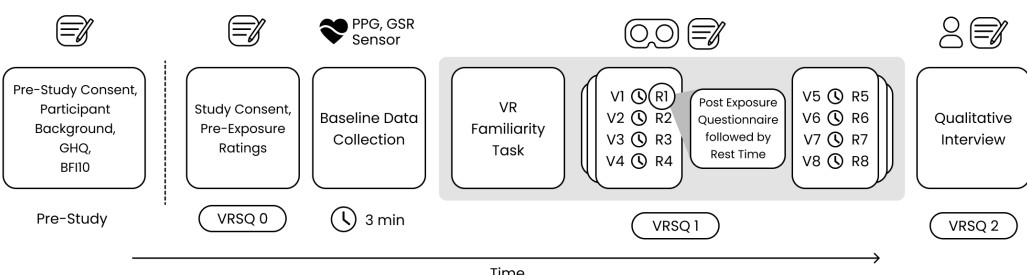

Figure 1: Illustration of our Experiment protocol for data collection.

1. **Pre-Study Survey:** The data collection is initiated by collecting participants' consent. Then, we collected participants' background details (gender, age, educational background, and prior VR exposure), personality scores using BFI-10 questionnaires, and information on prior psychological well-being using GHQ-12, a popular screening questionnaire designed to identify common psychiatric conditions within non-psychiatric clinical and research settings. We have used *over the past week* version of the GHQ-12 questionnaire to collect participants' psychological well-being before participating in our study. GHQ-12 helps mitigate the bias that might be introduced if the participant experienced some psychological

lows or highs prior to data collection. The GHQ-12 and personality scores were added as additional contextual details alongside other participant characteristics in our dataset.

2. **Pre-VR exposure:** Following the pre-study survey, participants were introduced to the experiment protocol and sensor setup along with instructions for data collection and the study's possible risks or discomfort (more details in section 4.1 of supplementary). Following instructions, participants **pre-exposure ratings** were gathered. This included PANAS, SAM, and Virtual Reality Sickness Questionnaire (VRSQ) (Kim et al. (2018)) scales to collect participants' baseline emotions and pre-VR sickness and fatigue symptoms, if any. The PANAS scale was used to collect positive and negative affect readings on a 5-point scale with ten positive (Interested, Strong, Enthusiastic, Proud, Inspired, Determined, Alert, Attentive, Active) and ten negative (Distressed, Irritable, Guilty, Scared, Upset, Hostile, Jittery, Ashamed, Nervous, Afraid) emotions. The SAM scale was used to collect scores for emotions' valence, arousal, and dominance dimensions. Following baseline ratings, PPG and EDA sensors were attached to the participant's non-dominant hand fingers. Subsequently, participants were asked to relax for 3 minutes of **baseline data collection**. More details on scales are provided in supplementary section 4.4.

3. **VR Familiarity:** Following baseline, participants were familiarized with the VR environment to mitigate any bias that may arise due to VR familiarity by making them initially sit in a VR waiting room to acclimate to the technology, exploring the surroundings by looking around for approx 4 minutes. Following this, participants transitioned to familiarizing themselves with the VR controller by engaging in a simple game where they used the VR controller to pick up and throw a ball into a box within the VR room. More details on the VR module are provided in supplementary section 4.3.

4. **VR Stimulus Exposure**: Participants were instructed to choose the assigned playlist after the VR familiarity task. Each playlist contained eight videos retrieved from a public database of annotated 360° Videos (Li et al. (2017)). These videos were selected to elicit emotions from all four emotional quadrants of the Russell circumplex model (Russell et al. (1989)). The circumplex model organizes emotions based on two dimensions: valence and arousal. Thus, videos from four categories were shown to participants: **High Valence-High Arousal (HVHA)**: elicits high energy positive emotions (such as excitement, Joy) **High Valence-Low Arousal (HVLA)**: elicits low energy positive emotions (such as calmness, relaxation) **Low Valence-High Arousal (LVHA)**: elicits high energy negative emotions (such as stress, anger) **Low Valence-Low Arousal (LVLA)**: elicits low energy negative emotions (such as boredom, depression)

Following each VR video, *post-exposure ratings* were collected from subjects to annotate their emotions during the VR exposure. The post-exposure questionnaire was the same as the pre-exposure, with additional questions about familiarity and liking (Koelstra et al. (2012)) of content. The familiarity score was collected on a 1-5 scale. Between subsequent VR videos and self-reporting, participants were given rest periods to avoid VR sickness or fatigue. Additionally, participants were asked to fill *VRSQ* after completing the fourth and eighth VR videos. More details on stimulus selection, stimulus order, and playlist creation are provided in section 4.2 of the supplementary document.

5. **Qualitative Interview:** At the end of physiological data collection, the participant's sensors were removed following the qualitative interview. The semi-structured interviews allowed us to adapt the questions based on the participants' feedback. The objective of the interview was to prompt the participants to articulate the emotions that they experienced while watching the VR stimulus and the reason behind those emotions. We used a monitor to show the VR videos from the assigned playlist in order to support the participants in recalling the stimulus while explaining the emotions. The questions like, *"What was the major emotion felt in this video (referring to the video)?"* and *Were there any mixed emotions that you (participant) felt while being exposed to stimulus* were asked to capture the subjective experiences. The interview was audio-recorded after obtaining consent from participants. Later, the audio recording is converted into text during dataset preparation using Google speech-to-text API [2]. The data is then manually cleaned to extract each subject's response to the interviewer's questions.

---

[2] https://cloud.google.com/speech-to-text

## 3.2 Experimental Setup

EEVR consists of two physiological signals: Electrodermal Activity (EDA) and Photoplethysmography (PPG). The physiological signals are recorded using the *4-channel Biopac MP36* [3] system. The Biopac MP36 consists of 4 channels to collect a maximum of four synchronized signals simultaneously. The MP36 system was connected to BSL4 data acquisition software to visualize and store the physiological signal data and to the peripheral PPG and EDA sensors. The EDA sensor module (SS57LA Hardware module [4]) was attached to the index and middle fingers (Tabbaa et al. (2022)) of the participant's non-dominant hand, utilizing EL507 Electrodes for collecting users' skin electrical conductance. The PPG sensor module (SS4LA Hardware module [5]) was attached to the participant's non-dominant hand ring finger. Non-dominant was used to attach sensors for minimizing noise due to motion artefacts. Before attaching the sensors, Isotonic Gel was applied to EDA electrodes to ensure minimal noise in the collected data. The biopac MP36 has been used in prior research for collecting physiological signal data (Shafiq and Veluvolu (2017); Zalabarria et al. (2020); Aeimpreeda et al. (2020); Maheshkumar et al. (2016)). It has a resolution of 2000 Hz for all acquired physiological signals. For 360° video stimulus, a *Meta Quest Pro* headset was used. This headset has 2 x LCD panels with 1800 x 1920 pixels per eye, a refresh rate of 90Hz, and a 106º Horizontal × 96º Vertical Field of view. It incorporates eye relief adjustment, lens spacing, and spatial audio support. We have used the OpenXR plugin to integrate the Meta Quest pro headset with the Unity application. OpenXR plugin also helped us with hand gestures and controls for interacting with the application's user interface. Pre and post-exposure ratings were collected using iPad Pro Tablet.

## 3.3 Participants and Experiment Details

EEVR comprised 37 healthy participants (21 males, 16 females) aged 18-33 (M=23.1, SD=4.02). Participants were from varying educational backgrounds - Bachelor (24), Master (8), Senior High School (4) and Doctorate (3). Our exclusion criteria exclude individuals with experience or a history of heart issues, heart arrhythmia, high blood pressure, medical conditions affecting equilibrium, visual or auditory impairments, neurological ailments, cognitive challenges, psychological issues, or diagnosed depression, as per the guidelines laid out in (Tabbaa et al. (2022)). Additionally, participants with low proficiency in the English language were not included in the study to avoid any impact of language understanding on the participants. All participants included in the study were requested to sign the consent form. Participants were also instructed to forbid any caffeine intake and refrain from exercising 3 hours before the experiment. The study was conducted in an institute research lab with minimal disturbance. The experiment setup (room temperature and sitting arrangements) remained the same for all the participants. The experiment was conducted with the experimenters present in the lab.

## 3.4 Dataset Description

The EEVR dataset comprises 296 emotion tasks plus 37 baselines, with each of the 37 participants experiencing eight VR 360° videos. These tasks aim to gather physiological data, totaling approximately 797 minutes and 83 seconds, including each participant contributing 3 minutes of baseline data. Along with physiological data, 296 textual descriptions were collected from 37 participants through interviews. The physiological data segment was identified with video_ID and subject_ID. The EDA and PPG data were originally collected at a sampling frequency 2000Hz but were downsampled to 15.625Hz for EDA and 125Hz for PPG data. The downsampling was done to reduce computation costs while maintaining the data quality. Prior work has utilized EDA data at a sampling frequency of 4Hz and PPG of 64Hz minimum (Schmidt et al. (2018)). More details about dataset preparation, cleaning, and analysis are provided in the supplementary section 5.

## 3.5 Annotation

All physiological data segments (Participant_ID - Video_ID) are annotated with self-reported ratings of arousal, valence, dominance (using SAM scale), discrete emotional ratings using PANAS (further used to calculate positive and negative affect scores), and additionally, we have a qualitative textual description for each data segment. Moreover, liking and familiarity scores on a scale of 1-5 are

---

[3] https://www.biopac.com/product/mp36r-systems/

[4] https://www.biopac.com/product/eda-lead-bsl/

[5] https://www.biopac.com/product/photoplethysmogram-for-pulse-waveform-bsl/

also present for each segment. Further, we have personality scores and GHQ-12 ratings for each participant. More details on the affect score and GHQ score calculation and annotation analysis have been added to the supplementary sections 4.4 and 5.1.

**Labels for Supervised Learning**: For supervised learning, we propose three 2-class labels based on both participants' responses to arousal, valence questionnaire, and based on stimulus annotations. The arousal data was collected on a scale of 1-5, which was further divided into binary classes by considering data with 0-3 ratings as low arousal and 4-5 as high arousal; we followed a similar process for categorizing valence data. The physiological data collected during video stimulus from LVHA, LVLA, and baseline are annotated as negative emotions, while the videos from HVHA and HVLA are annotated as positive, creating binary classes. The baseline was annotated as negative valence, considering the stress that participants may undergo due to the sensor attachment procedure and activities before the experiment. Upon analysis, we found our arousal labels were skewed compared to other labels. To overcome this skewness, we have used oversampling for the arousal labels.

## 4 Experiments

### 4.1 Baseline

We conducted baseline classification for three tasks: Arousal Classification, Valence Classification, and Stimulus-label-based Emotion Classification. Each task involved binary labels. We have performed baseline classification separately for each data modality: EDA and PPG for all three labels. Followed by multimodal classification of physiological signals combining EDA and PPG data. All physiological signal-based baseline experiments were conducted using Leave-one-subject-out (LOSO) cross-validation (Saganowski et al. (2023)). All the results are presented in Table 2 as the average performance across all LOSO subjects, calculated over three different seed values. To validate our text data, we also performed baseline classification tasks for only text data, and the results are presented in Table 2. Next, we conducted contrastive training (Radford et al. (2021)) to present our pre-training method using the paired physiological signals and textual data. The baseline results with or without contrastive training on 296 (excluding baseline samples) text-physiological signal pairs are presented in Table 3. The code for all baseline implementations is present here `https://github.com/alchemy18/EEVR/`. Next, we present more details on physiological signals, text, and contrastive baseline.

| Modality | Models | Stimulus-label | | Valence | | Arousal | |
|---|---|---|---|---|---|---|---|
| | | Accuracy | F1 Score | Accuracy | F1 Score | Accuracy | F1 Score |
| EDA | **Logistic Regression** | 86.78 ± 0 | 0.82 ± 0 | 61.56 ± 0 | 0.71 ± 0 | 47.41 ± 0 | 0.36 ± 0 |
| | **Decision Tree** | 85.09 ± 0.17 | 0.83 ± 0 | 58.46 ± 1.06 | 0.64 ± 0.01 | 54.05 ± 1.12 | 0.35 ± 0.02 |
| | **Random Forest** | **90.79 ± 0.46** | **0.89 ± 0.01** | 60.26 ± 1.81 | 0.66 ± 0.01 | 57.23 ± 1.19 | 0.28 ± 0.04 |
| | **LDA** | 87.69 ± 0 | 0.85 ± 0 | **61.86 ± 0** | **0.69 ± 0** | 48.97 ± 0 | 0.37 ± 0 |
| | **XGBoost** | 90.69 ± 0.52 | 0.89 ± 0.01 | 59.76 ± 0.52 | 0.66 ± 0.01 | 56.61 ± 0.34 | 0.37 ± 0.01 |
| | **SVM** | 85.29 ± 0 | 0.81 ± 0 | 59.16 ± 0 | 0.71 ± 0 | 51.66 ± 0 | **0.44 ± 0** |
| | **MLP** | 87.39 ± 0 | 0.85 ± 0 | 61.86 ± 0 | 0.68 ± 0 | **57.27 ± 0** | 0.39 ± 0 |
| PPG | **Logistic Regression** | 81.08 ± 0 | 0.77 ± 0 | 61.26 ± 0 | 0.70 ± 0 | **56.29 ± 0** | **0.42 ± 0** |
| | **Decision Tree** | 68.87 ± 0.35 | 0.65 ± 0 | 54.35 ± 0.30 | 0.59 ± 0.01 | 49.43 ± 0.32 | 0.26 ± 0.01 |
| | **Random Forest** | 75.88 ± 0.35 | 0.69 ± 0.01 | **61.66 ± 1.93** | **0.70 ± 0** | 49.27 ± 0.42 | 0.18 ± 0.01 |
| | **LDA** | **81.08 ± 0** | **0.78 ± 0** | 58.96 ± 1.73 | 0.67 ± 0.06 | 54.47 ± 3.72 | 0.40 ± 0.02 |
| | **XGBoost** | 49.44 ± 0 | 0.68 ± 0 | 57.26 ± 0.76 | 0.64 ± 0.01 | 47.89 ± 7.57 | 0.26 ± 0.13 |
| | **SVM** | 80.48 ± 0 | 0.75 ± 0 | 59.86 ± 1.91 | 0.70 ± 0.05 | 47.99 ± 3.78 | 0.32 ± 0.10 |
| | **MLP** | 78.68 ± 0 | 0.75 ± 0 | 56.76 ± 0 | 0.66 ± 0 | 54.16 ± 0 | 0.38 ± 0 |
| PPG + EDA | **Logistic Regression** | 85.89 ± 0 | 0.82 ± 0 | 60.06 ± 0 | 0.69 ± 0 | 55.23 ± 0 | 0.41 ± 0 |
| | **Decision Tree** | 83.78 ± 0.80 | 0.83 ± 0.01 | **62.77 ± 0.30** | 0.66 ± 0 | **58.13 ± 0.70** | 0.40 ± 0.01 |
| | **Random Forest** | **90.69 ± 0** | **0.89 ± 0** | 61.06 ± 1.35 | 0.70 ± 0.01 | 56.78 ± 1.56 | 0.26 ± 0.01 |
| | **LDA** | 84.89 ± 1.39 | 0.82 ± 0.01 | 57.56 ± 2.95 | 0.66 ± 0.06 | 55.48 ± 1.04 | **0.42 ± 0.01** |
| | **XGBoost** | 87.19 ± 2.73 | 0.85 ± 0.03 | 61.36 ± 4.79 | 0.67 ± 0.04 | 58.0 ± 1.66 | 0.36 ± 0.06 |
| | **SVM** | 87.29 ± 1.39 | 0.84 ± 0.02 | 62.16 ± 2.08 | **0.72 ± 0.02** | 55.97 ± 3.44 | 0.38 ± 0.04 |
| | **MLP** | 83.48 ± 0 | 0.81 ± 0 | 58.86 ± 0 | 0.63 ± 0 | 56.89 ± 1.47 | 0.36 ± 0.03 |
| Text | **DistillBert** | **97.44 ± 0.69** | **0.97 ± 0.01** | **91.73 ± 1.73** | **0.88 ± 0.02** | **89.94 ± 1.17** | **0.88 ± 0.02** |
| | **XLMBert-a Base** | 97.32 ± 0.34 | 0.97 ± 0 | 89.46 ± 1.60 | 0.70 ± 0.15 | 76.50 ± 9.59 | 0.70 ± 0.15 |

Table 2: Results for Arousal Classification, Valence Classification, and Stimulus-label-based Emotion Classification on EDA, PPG and Text Data

### 4.1.1 Physiological Signal Baseline

We present our baseline model using hand-crafted features from our physiological signal data, similar to prior work on emotion recognition (Schmidt et al. (2018); Shukla et al. (2021); Ninh et al. (2022); Chaptoukaev et al. (2024)). To extract our features, we performed the following steps: First, each EDA and PPG data segment is filtered. EDA data is filtered using a low-pass filter with a 5Hz cutoff frequency and a 4th-order Butterworth filter, while PPG data is cleaned using a bandpass filter. Next, the EDA data is decomposed into tonic and phasic components, referred to as skin conductance level (SCL) and skin conductance response (SCR) using cvxEDA, a convex optimization-based approach (Greco et al. (2015)). We then extracted statistical features, including dynamic range and slope, from both SCR and SCL components and time-domain features from SCR, such as the number of peaks, average amplitude, and duration. For PPG data, feature extraction is performed using the Neurokit Library (Makowski et al. (2021)) to extract HRV-related time domain, frequency domain, and non-linear features. After feature extraction, the features are normalized participant-wise using min-max scaling. Classical machine learning algorithms are then applied to the extracted features for all three classification tasks. We combine the handcrafted features from both modalities to train our multimodal machine-learning models using both EDA and PPG data. All machine learning models are trained using default hyperparameters from sklearn. For the training MLP, we used two hidden layers with 50 and 100 dimensions. All classification models are trained using the following seeds: 42, 43, and 111. Results are presented in table 2 and additional details about experiments are provided in supplementary section 5.5.

### 4.1.2 Textual Data Baseline

Next, we performed the three classification tasks on our textual data (for 296 samples excluding baseline). We followed the standard text classification pipeline, starting with data preprocessing, which includes data cleaning (removing stopwords and punctuation, converting to lowercase) and lemmatization. Following this, we applied tokenization, and then we fine-tuned two pre-trained models (DistilBERT (Sanh et al. (2019)) and XLM-RoBERTa Base (Conneau et al. (2019))) for the classification tasks. For our experiments, we have generated five random splits, with 80% of the data used for training and 20% for testing. Results are presented in table 2.

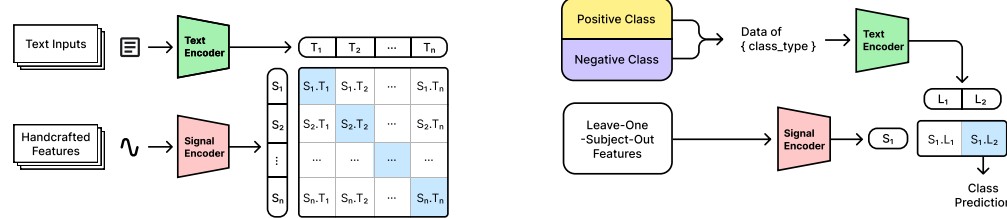

Figure 2: The Architecture for Contrastive-Language Singal Pre-Training (CLSP).

### 4.2 Contrastive Language-Signal Pre-training

To underscore the importance of integrating textual descriptions in emotion recognition, we introduce the Contrastive Language-Signal Pre-training (CLSP) method for extracting more contextualized representations. The model was trained on physiological signals and text pairs to learn a joint embedding space, where both modalities are closely aligned using a contrastive loss function Radford et al. (2021). Following pre-training, we evaluated the model's performance on test subject data using the leave-one-subject-out cross-validation approach, leveraging minimal labels generated in the format "Data of {class_Type}" (e.g., "Data of positive emotion class"). CLSP employs separate neural networks to process the handcrafted features of physiological signals (PPG and EDA signal data) and text data. For signal data, linear layers with hidden dimensions of 50 and 100 are utilized, while the text data is processed using a pre-trained DistilBERT (transformer-based language model). These extracted feature representations are then used to optimize a contrastive objective, maximizing the similarity between positive pairs and minimizing it for negative pairs. The detailed architecture is depicted in Figure 2, and our results for CLSP are summarized in table 3. We found that the emotion recognition for arousal and valence tasks using the CLSP method led to significant improvement in

classification results compared to without-text-supervision (Hand-crafted features + Neural Network (two linear layers of dimensions 50, 100)) training. This highlights the effectiveness of incorporating qualitative textual descriptions into physiological signal-based emotion representation learning. Further experimental details and comprehensive discussions are provided in Supplementary Section 5.5.

| Modality | Model | Stimulus-label | | Valence | | Arousal | |
|---|---|---|---|---|---|---|---|
| | | Accuracy | F1 Score | Accuracy | F1 Score | Accuracy | F1 Score |
| **EDA** | **HC+NN** | **87.39** | **0.85** | 61.86 | 0.68 | 57.27 | 0.39 |
| **PPG** | **HC+NN** | 78.68 | 0.75 | 56.76 | 0.66 | 54.16 | 0.38 |
| **EDA+PPG** | **HC+NN** | 83.48 | 0.81 | 58.86 | 0.63 | 58.58 | 0.40 |
| **EDA+Text** | **CLSP** | 64.19 | 0.68 | **70.38** | **0.73** | **77.25** | **0.81** |
| **PPG+Text** | **CLSP** | 56.95 | 0.53 | 64.74 | 0.64 | 69.91 | 0.62 |
| **EDA+PPG+Text** | **CLSP** | 53.50 | 0.48 | 64.87 | 0.60 | 69.64 | 0.64 |

Table 3: Results for Physiological Baseline without text using Hand-crafted features + NN and with text using CLSP on 296 text-signal pairs for seed=43 and epoch=15.

## 4.3 Zero-Shot Transfer

To assess the generalization capabilities of our pre-trained CLSP models across datasets collected in varied environments, we conducted a zero-shot transferability evaluation on our pre-trained model. For these experiments, we utilized three datasets representing distinct data collection settings: Emognition, acquired using 2D video stimuli in laboratory settings (Saganowski et al. (2022)), WESAD, gathered using the TSST psychological task and video stimuli within controlled lab conditions (Schmidt et al. (2018)), and NURSE, recorded in real-life hospital environments during the COVID-19 pandemic (Hosseini et al. (2022)). As detailed in Table 4, our pre-trained model demonstrated the ability to predict emotions in these new domains with accuracy comparable to the baseline models, and in several instances, it even surpassed the performance of supervised baselines. These findings show the effectiveness of integrating text-based emotion descriptions for learning representations that transfer robustly across diverse data domains, irrespective of the environment, device, or participant demographics. To ensure fair comparisons, we employed a standardized pipeline encompassing data cleaning, participant-wise normalization, feature extraction, and classification across all experiments.

| Dataset (Signal Type) | Method | Arousal | | Valence | |
|---|---|---|---|---|---|
| | | Accuracy | F1 Score | Accuracy | F1 Score |
| **Emognition (EDA)** | **MLP** | 52.80 | 0.57 | **61.89** | 0.36 |
| | **Zero-shot CLSP** | **53.23** | **0.59** | 50.32 | **0.49** |
| **Emognition (PPG)** | **MLP** | 49.94 | 0.53 | 50.63 | 0.28 |
| | **Zero-shot CLSP** | 48.19 | 0.47 | 51.88 | 0.41 |
| **Emognition (EDA + PPG)** | **MLP** | 51.53 | 0.54 | 55.12 | 0.34 |
| | **Zero-shot CLSP** | 50.94 | 0.52 | 53.58 | 0.41 |
| **WESAD (EDA)** | **MLP** | 85.00 | 0.84 | 96.67 | 0.97 |
| | **Zero-shot CLSP** | 53.33 | 0.67 | 51.67 | 0.67 |
| **WESAD (PPG)** | **MLP** | 80.00 | 0.80 | 75.00 | 0.75 |
| | **Zero-shot CLSP** | 70.00 | 0.68 | 66.67 | 0.72 |
| **WESAD (EDA + PPG)** | **MLP** | **91.67** | **0.91** | **98.33** | **0.98** |
| | **Zero-shot CLSP** | 75.00 | 0.71 | 86.67 | 0.86 |
| **Nurse (EDA)** | **MLP** | 39.88 | 0.32 | 71.83 | 0.03 |
| | **Zero-shot CLSP** | **55.48** | **0.58** | **84.93** | 0.20 |
| **Nurse (PPG)** | **MLP** | 45.10 | 0.38 | 72.08 | 0.05 |
| | **Zero-shot CLSP** | 53.08 | 0.48 | 75.34 | 0.23 |
| **Nurse (EDA + PPG)** | **MLP** | 48.35 | 0.43 | 76.04 | 0.23 |
| | **Zero-shot CLSP** | 53.08 | 0.45 | 84.59 | **0.42** |

Table 4: Zero-shot transferability results of our pre-trained model (CLSP) compared to supervised baseline model trained on existing datasets (Emognition, WESAD, and Nurse)

# 5 Limitations

*EEVR* dataset is collected using pre-annotated virtual reality videos within controlled laboratory settings. The experiment design does not consider the influence of external factors that may impact participants' emotional responses to immersive stimuli and assumes isolated responses to stimuli. Factors such as VR sickness, familiarity with VR technology, and attitudes toward this new technology may also affect participants' emotional responses. Furthermore, the placement of sensors and the VR headset can cause discomfort. Therefore, the signals recorded in this setting may not necessarily replicate real-life responses from all participants. Moreover, training based on participants' ratings is susceptible to participant bias, which may affect subsequent results. To address the subjectivity of emotional responses, we have collected qualitative responses in the form of textual descriptions, providing rich contextual annotations alongside objective ratings. Additionally, *EEVR* contains data from a privileged set (upper middle class, educated) of the audience and does not represent other sections of society and thus is biased towards a specific society group.

# 6 Ethical Considerations and Dataset Accessibility

*EEVR* study is approved by the Institution review board [6] of IIIT-Delhi registered with the National Ethics Committee Registry for Biomedical and Health Research (NECRBHR). All participants in this study provided explicit consent for recording their physiological signals and audio data during qualitative interviews and for releasing this data for research purposes. To protect their identities, participants were pseudonymized using numerical identifiers. The audio data was transcribed, manually checked for any identifying information, and included as textual descriptions devoid of sensitive content. Participants received merchandise goodies worth 5.39 USD for their participation. The dataset is available for download under a CC BY-NC-SA license for non-commercial research purposes on our website. The codes for data cleaning, feature extraction, and classification are open-source and can be accessed. The open-source code can be accessed through the following repository. Our dataset does not have any direct negative impact on society and is designed and made open source, keeping users' privacy in mind.

# 7 Conclusion

Understanding emotions is vital for effectively addressing mental health challenges. While self-reports are considered the optimal method for collecting emotional data, they often use objective scales that can oversimplify emotions, which are inherently continuous, resulting in the loss of valuable information. In EEVR, we stress the importance of collecting elaborate self-reports in the form of text or audio data. This kind of self-report provides rich contextual information that reveals better correlations between physiological signals and emotions. Our experimental protocol is easily replicable in lab settings and can be extended to daily-life data collection through chatbots and audio input-based companion applications. The EEVR dataset is a valuable resource for machine learning researchers working with physiological data, enabling them to build upon existing baselines or develop their own algorithms. Pairing physiological signals with textual descriptions facilitates the development of advanced emotion recognition algorithms using wearable devices. This dataset enhances understanding of emotions and their correlations with participants' behavioral characteristics, such as psychological well-being, personality, and physiological changes. Additionally, the open-source baseline codes and easy accessibility of the EEVR dataset support future reproducibility and encourage new initiatives leveraging this dataset.

# 8 Acknowledgements

The authors gratefully acknowledge the support of the iHub-Anubhuti-IIITD Foundation, established under the NM-ICPS scheme of the DST, and the Center of Excellence in Healthcare (CoEHe) at IIIT-Delhi. We also extend our sincere gratitude to our participants for their willingness to participate and share their data with us. Thanks to Akshit Jindal for his valuable support in peer-reviewing the paper. Finally, we acknowledge the support of the Microsoft Research Travel Grant, which enabled us to present this work at NeurIPS 2024.

---

[6] https://irb.iiitd.edu.in/

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
