# EEVR: A Dataset of Paired Physiological Signals and Textual Descriptions for Joint Emotion Representation Learning

Pragya Singh[1], Ritvik Budhiraja[1], Ankush Gupta[1], Anshul Goswami[1], Mohan Kumar[2], and Pushpendra Singh[1]

[1]IIIT-D, New Delhi, India , `pragyas@iiitd.ac.in`, `ritvik19322@iiitd.ac.in`, `ankush21232@iiitd.ac.in`, `anshul20361@iiitd.ac.in`, `psingh@iiitd.ac.in`
[2]RIT, Rochester, New York, US , `mjkvcs@rit.edu`

## Supplementary Materials

38th Conference on Neural Information Processing Systems (NeurIPS 2024) Track on Datasets and Benchmarks.

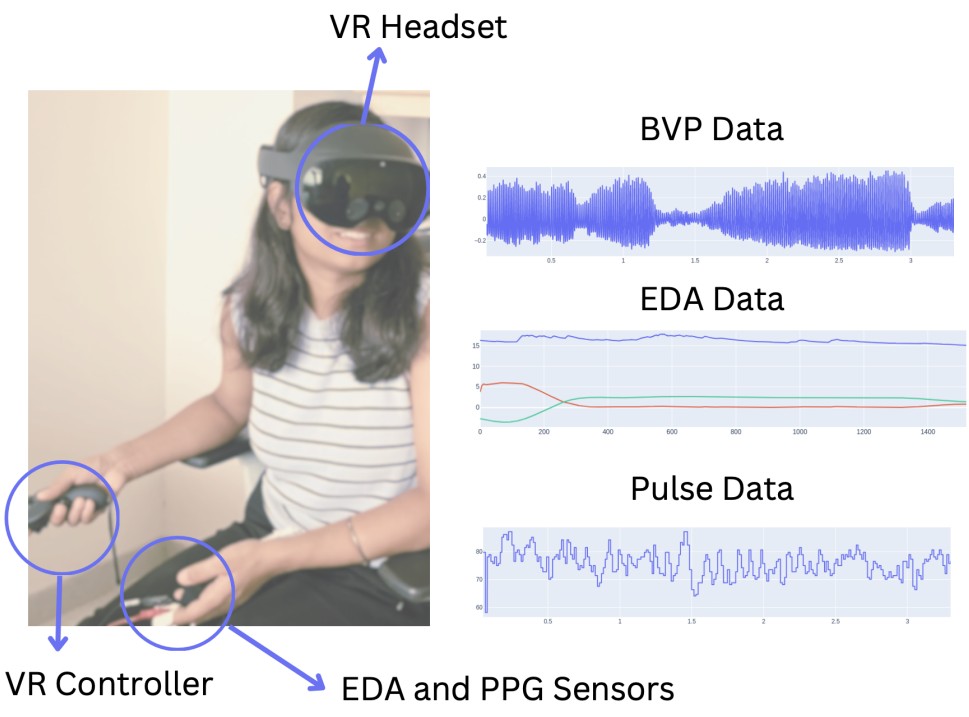

Figure 1: Experiment Setup of EEVR dataset

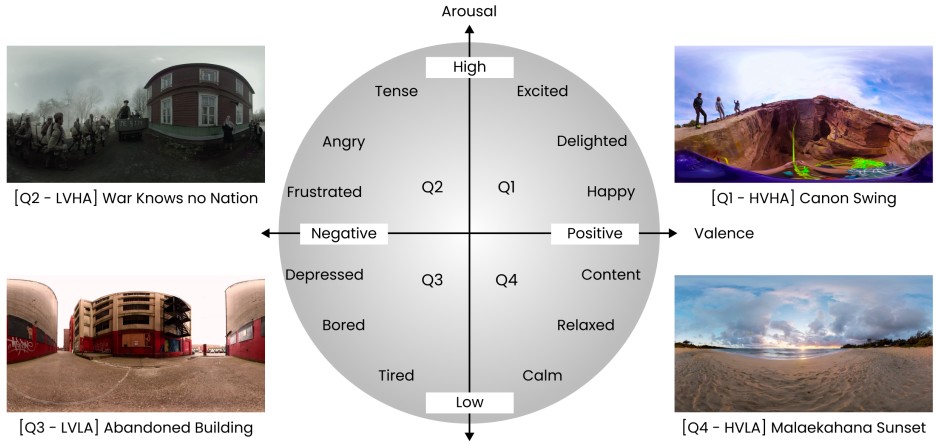

Figure 2: The figure presents still images extracted from 360° videos used in the experiment to display various environments to the participants. These images represent different valence-arousal combinations, including Q1: High-Valence-High-Arousal (HVHA), Q2: Low-Valence-High-Arousal (LVHA), and Q3: Low-Valence-Low-Arousal (LVLA), Q4: High-Valence-Low-Arousal (HVLA). The videos were selected from the publically available 360° VR video dataset (Li et al. (2017).)

# 1 EEVR Overview

The *EEVR* dataset comprises synchronized pairs of physiological signals and textual data. It includes responses to four self-assessment questions regarding perceived arousal, valence, dominance, and discrete emotions ratings collected using PANAS questionnaires (which were further utilized to calculate Positive and Negative Affect Score). The EEVR dataset was collected using Virtual Reality (VR) 360° videos as the elicitation medium. The videos utilized in the dataset were selected based on their arousal and valence ratings to cover all four quadrants of the Russell circumplex emotion model (Russell et al. (1989)), as shown in Figure 2. The remainder of the supplementary materials provide detailed information about the *EEVR* dataset. Figure 3 provides a datasheet for the EEVR dataset based on Gebru et al. (2018). The experiment setup is presented in Figure 1.

## 1.1 EEVR Size Details

The EEVR dataset consists of data from 37 healthy participants who agreed to share their data publicly. Although 41 participants were involved in the data collection study, data from only 37 participants is publicly available. During data acquisition, the data from four participants was damaged due to factors like motion sickness in the VR environment and issues with sensor attachment. Consequently, the dataset provides physiological signal data (including Electrodermal Activity (EDA) and Photoplethysmogram (PPG) signals) and textual descriptions of emotions felt during each emotional stimulus for 37 participants.

The EEVR dataset comprises 296 tasks in total, with each participant experiencing eight VR 360° videos shown to induce emotions from all four quadrants of the Russell emotion model (two videos from each quadrant). Table 1 presents a summary of the minute durations for each video, along with their respective playlist details. Further details regarding the playlist and video order are elaborated in Section 4.2.

| Video Name | Count (minutes) | Playlist |
|---|---|---|
| The Displaced | 3:23 | 1, 3 |
| Happyland 360 | 2:43 | 1, 3 |
| Jailbreak 360 | 3:06 | 1, 3 |
| War Knows No Nation | 3:15 | 1, 3 |
| Canyon Swing | 1:44 | 1, 3 |
| Redwoods Walk Among Giants | 2:00 | 1, 3 |
| Speed Flying | 2:34 | 1, 3 |
| Instant Caribbean Vacation | 2:30 | 1, 3 |
| The Nepal Earthquake Aftermath | 3:09 | 2, 4 |
| Zombie Apocalypse Horror | 3:00 | 2, 4 |
| Abandoned building | 3:00 | 2, 4 |
| Kidnapped | 2:58 | 2, 4 |
| Mega Coaster | 1:57 | 2, 4 |
| Malaekahana Sunrise | 3:29 | 2, 4 |
| Puppies host SourceFed for a day | 1:20 | 2, 4 |
| Great Ocean Road | 1:58 | 2, 4 |

Table 1: Video names with their duration details and the playlist number

## 1.2 EEVR Organization and File formats

The EEVR dataset, as downloaded, is organized into two main subdirectories, as illustrated in Figure 4. The first subdirectory contains processed physiological data, including CSV files with raw EDA and PPG data, organized by participant details and Video ID for ease of use. It also includes EDA and PPG features CSV files extracted using the feature extraction pipeline. The second subdirectory holds raw data and is divided into four playlist folders. Each playlist folder contains directories for subject-wise raw EDA text files, raw PPG text files, annotation text files, and raw ACQ files in the original Biopac format. All the physiological data files are organized in .CSV and .TXT formats, making them easily usable for all programming languages. All physiological signals were initially sampled at 2000Hz but were downsampled to 128Hz for PPG and 15.68Hz for EDA to reduce computational requirements. The .ACQ files contain the original 2000Hz data, while the .TXT and .CSV files are the downsampled versions used for experimentation. The downsampling frequencies

# EEVR Dataset Facts

**Dataset** EEVR

### Motivation

**Summary** EEVR is a multimodal dataset designed for emotion recognition. It comprises physiological signal data collected from wearable sensors along with raw textual captions corresponding to each emotion elicitation segment.
**Example Use Case** Emotion Recognition, Arousal classification, Valence classification, Personality Recognition
**Original Authors** P. Singh, R. Budhiraja, A. Gupta, A. Goswami, M. Kumar, P. Singh

### MetaData

| | |
|---|---|
| **URL** | https://melangelabiiitd.github.io/EEVR/ |
| **KeyWords** | Emotion Recognition, Wearable sensor, Physiological signal |
| **Format** | .acq, .csv, .txt |
| **Ethical Review Approval** | IRB-IIIT-Delhi, NECRBHR |
| **License** | CC BY-NC-SA |
| **First Release Year** | 2024 |

### Sensors

| | |
|---|---|
| **EDA** | SS57LA, 4-channel Biopac MP36 |
| **PPG** | SS4LA, 4-channel Biopac MP36 |

### Data Annotation

| | |
|---|---|
| **Self-Assessments** | Arousal, Valence, Dominance, Positive Affect Score, Negative Affect Score, Familiarity, Discrete Emotions |
| **Textual Labels** | Raw Textual Description per video stimuli |
| **Additional Data** | Personality Score (BFI10), GHQ, VRSQ |

### Participants

| | |
|---|---|
| **Count** | 37 |
| **Gender** | 21 males, 16 females) |
| **Age** | 18-33 (M=23.1, SD=4.02) |
| **Backround** | Bachelor's (24), Master's (8), Senior High School (4), Doctorate (3) |

### Dataset Size

| | |
|---|---|
| **Total Size** | 668 MB |
| **Physiological Data Duration** | 797 minutes and 83 seconds |

Figure 3: Dataset Summary card for EEVR, constructed based on (Gebru et al. (2018))

were chosen based on previous research to ensure no information was lost. Participant 29's data was collected in two parts due to a disconnection during the experiment, resulting in two raw data files.

Additionally, there are three files: one detailing participant information collected during the data collection process, second with self-assessment details collected during data collection, and third with text data from semi-structured interviews for each participant-video segment. We have also included VR_Application.zip file containing the VR environment simulation build file and video resources. The Participant_details is organized into sheets for Participant Details, GHQ-12, and BFI-10. The Self_Assessment file is further organized into sheets for the Pre-exposure Questionnaire, Post-exposure Questionnaire, Affect-Personality Score, and VRSQ Scores. The text data file also contains sheets for Text-Labels (Text description with information on participant ID and video ID and labels), Video description, and Video ID and Video Name mapping information.

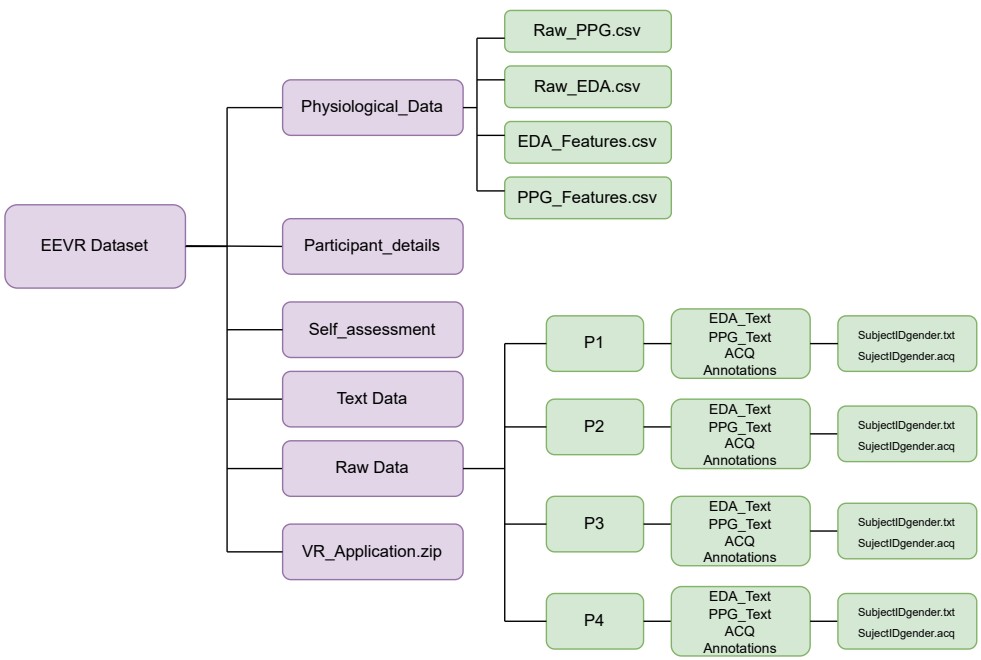

Figure 4: File Organization of EEVR dataset

# 2 EEVR Usage and Publishing

## 2.1 Intended Uses

The *EEVR* dataset is collected and published to further the research in Emotion recognition using physiological signal data. The dataset is a resource for synchronised physiological signals, a textual description of emotions felt and annotations in the form of perceived self-reports. Several use cases, including extracting emotions from each modality and analyzing the correlations between physiological signals and various emotion labels. Further, the dataset can be used for pre-training physiological signal-based models using contrastive techniques for zero-shot classification of various tasks like Valence classification, Arousal classification and stimuli-label-based emotion classification.

## 2.2 Ethical Consideration

We acknowledge that, despite all precautions, there is a possibility of the dataset being misused by malicious users. The authors take full responsibility for any rights violations that may occur during the data collection process or any related work. They are committed to taking necessary actions, such as removing data that poses such issues, to address any problems that arise.

## 2.3 EEVR Licensing, Hosting, and Maintenance Plan

The EEVR dataset and its relevant code file are available for researchers to use further. The dataset is available to use under CC BY-NC-SA license for non-commercial research. It can be accessed by filling in the Dataset Access Request Form on our website. Upon completing the form, users can access the EEVR dataset stored on Google Drive. The authors will maintain the dataset files for the long term, ensuring that the file structures remain unchanged. The EEVR website will also be maintained for the long term, providing users with easy access to download the dataset. All the code files are available under the MIT open-source licence on github.

# 3    Human Subjects Considerations

The EEVR dataset collection study has been approved by the Institution review board [1] of IIIT-Delhi registered with the National Ethics Committee Registry for Biomedical and Health Research (NECRBHR). The participants for this study were recruited through email invitations. Before data collection began, all participants were introduced to the study protocol and its purpose. They were also informed about privacy concerns and any risks involved in the study. All the subjects participated on a voluntary basis. Additionally, all participants are made aware of our exclusion criteria. No participants with experience or a history of heart issues, heart arrhythmia, high blood pressure, medical conditions affecting equilibrium, visual or auditory impairments, neurological ailments, cognitive challenges, psychological issues, or diagnosed depression were recruited for this study. Further, participants with motion sickness issues were also excluded to avoid VR sickness discomfort on our subjects. The Ag/AgCl electrodes [2] used in our study have been proven in the past to adhere well to various types of skin surfaces. Further, we informed all participants to stop the experiments if they felt any discomfort. All the participant's data is pseudo-anonymized before being made publicly available.

# 4    Data Collection Protocol

## 4.1    Experiment Instruction and Sensors Preparation

Before starting the data collection, all participants were asked to sit comfortably in a chair. They were then informed about the study's purpose, which was to collect physiological data related to various emotions using VR 360° videos. Participants were instructed to report any discomfort or issues during the data collection and were informed to stop the experiment at any time in case of discomfort. The use of sensors and VR headsets was explained, and participants were given time to ask questions and express any concerns. Consent was then obtained from each participant. The preparation of wearable sensors involved attaching EDA (SS57LA) and PPG (SS4LA) sensor modules to the Biopac MP36 acquisition system. EL507 electrodes were prepared with isotonic gel and attached to the participants' index and middle fingers, while the PPG module was attached to the ring fingers. The EDA sensor module was calibrated by removing and reattaching one of the sensor heads. Following this, the sensors were checked for accurate readings. Upon confirmation of acquisition without any error, the experiment is started.

## 4.2    Stimulus Selection and Playlist Preparation

For this experiment, a total of 16 videos were selected from a publicly available 360° VR dataset containing 73 videos Li et al. (2017). To curate this subset, we applied a heuristic protocol based on the circumplex model of emotions Russell et al. (1989). We chose four videos from each category of the circumplex model, selecting those with the maximum distance from the origin to ensure a diverse and comprehensive representation. The 16 videos were then divided into two subgroups of eight videos each, as mentioned in Table 2. This decision was based on feedback from a pilot study (participants not included in the main study), the total experiment duration, and considerations to prevent participant fatigue or motion sickness from VR exposure. Alternate videos from each quadrant were paired to create two balanced video sets, ensuring normalization between the subgroups and a balanced experimental setting. After dividing the videos into subgroups, they were arranged in two different orders. In the first order, videos were organized based on their valence ratings, representing the degree of pleasantness or unpleasantness associated with an emotional state. The videos were arranged randomly in the second order, creating four playlists. The sorting technique employed for the study involved arranging videos from low negative valence to high positive valence. The four playlists are as follows- *Playlist1: VideoSet1 - Random Order*, *Playlist2: VideoSet1 - Valence Sorted Order*, *Playlist3: VideoSet2 - Random Order*, and *Playlist4: VideoSet2 - Valence Sorted Order*. The playlists are shown in Table 3. Participants were allocated playlists in a gender-balanced manner through random assignment. The valence-sorted orders were designed to induce emotions to transition smoothly between emotions, starting from positive emotions, then introducing more

---

[1] https://irb.iiitd.edu.in/
[2] https://www.biopac.com/product/eda-electrodes/

intense emotions, and finally transitioning to negative emotions. The random order was inspired by prior works that randomly showed videos.

| CMA Quadrant | VideoSet1 [V,A] | VideoSet2 [V,A] |
|---|---|---|
| HVHA | Canyon Swing [5.38, 6.88], Speed Flying [6.75, 7.42] | Mega Coaster [6.17, 7.17], Puppies host SourceFed for a day [7.47, 5.35] |
| HVLA | Redwoods Walk Among Giants [5.79, 2.0], Malaekahana Sunrise [6.57, 1.57] | Instant Caribbean Vacation [7.2, 3.2], Great Ocean Road [7.77, 3.92] |
| LVHA | Jailbreak 360 [4.4, 6.7], Zombie Apocalypse Horror [3.2, 5.6] | War Knows No Nation [4.93, 6.07], Kidnapped [4.83, 5.25] |
| LVLA | The Displaced [2.18, 4.73], The Nepal Earthquake Aftermath [2.73, 3.8] | Happyland 360 [3.33, 3.4], Abandoned Building [4.39, 2.77] |

Table 2: Videos categorized based on Valence (V), Arousal (A) rating

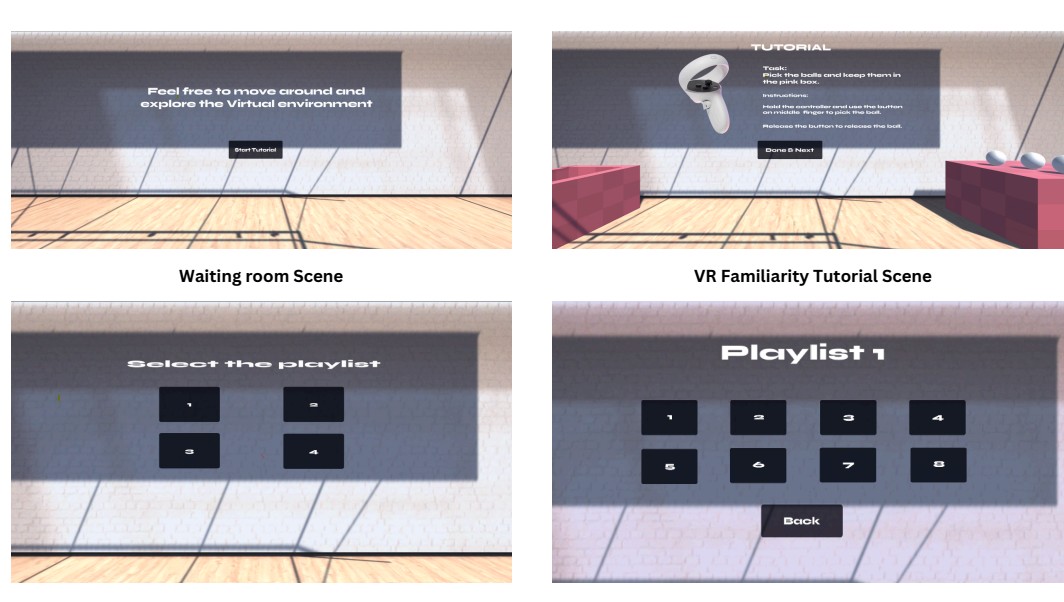

Figure 5: This figure illustrates the screens from Virtual Environment Room scene in following order: Waiting room scene, VR Familiarity Tutorial scene, Playlist Selection Scene and Video Selection Scene.

## 4.3 Virtual Reality Module Preparation

We developed a VR application for our experiment, enabling participants to experience emotionally stimulating videos. The application consists of two main components: 1) an introductory module to familiarize users with the VR environment and controllers, and 2) a video playback module for presenting 360° videos. The application was created using Unity. Since many users were new to VR, the introductory module starts with a "waiting room" scene designed to acclimate them to the VR environment. Instructions, including text and images, are displayed on the walls using Unity's XR UI canvas to guide users in interacting with and manipulating objects using the VR controller. To practice these skills, users complete a simple task of placing a ball in a bucket within the introductory scene. The second component allows users to experience 360° videos in VR. We curated four playlists, each containing eight videos, which were downloaded from a database[3] using the youtube-dl[4] tool.

[3]https://stanfordvr.com/360-video-database/
[4]https://github.com/ytdl-org/youtube-dl

| Video Name | Playlist ID-Video ID |
|---|---|
| The Displaced | P1V1 |
| Happyland 360 | P1V2 |
| Jailbreak 360 | P1V3 |
| War Knows No Nation | P1V4 |
| Canyon Swing | P1V5 |
| Redwoods Walk Among Giants | P1V6 |
| Speed Flying | P1V7 |
| Instant Caribbean Vacation | P1V8 |
| The Nepal Earthquake Aftermath | P2V1 |
| Zombie Apocalypse Horror | P2V2 |
| Abandoned Building | P2V3 |
| Kidnapped | P2V4 |
| Mega Coaster | P2V5 |
| Malaekahana Sunrise | P2V6 |
| Puppies host SourceFed for a day | P2V7 |
| Great Ocean Road | P2V8 |
| War Knows No Nation | P3V1 |
| Redwoods Walk Among Giants | P3V2 |
| Happyland 360 | P3V3 |
| Speed Flying | P3V4 |
| Instant Caribbean Vacation | P3V5 |
| Jailbreak 360 | P3V6 |
| The Displaced | P3V7 |
| Canyon Swing | P3V8 |
| Kidnapped | P4V1 |
| Malaekahana Sunrise | P4V2 |
| Zombie Apocalypse Horror | P4V3 |
| Puppies host SourceFed for a day | P4V4 |
| Great Ocean Road | P4V5 |
| Abandoned Building | P4V6 |
| The Nepal Earthquake Aftermath | P4V7 |
| Mega Coaster | P4V8 |

Table 3: Video Names and Video ID

These videos are in equirectangular panoramic format with a 3840 x 2160 pixels resolution. In Unity, separate scenes were created for each video, with texture renderers mapping the video frames to a skybox surrounding a central camera. A script tracks video playback in each scene, and once a video finishes, the user is returned to the playlist menu to select another video. This setup allowed us to collect users' physiological data for each video.

## 4.4 Self-Assessment

Each task in our study is annotated using Valence, Arousal, and Dominance. Additional data on liking and familiarity was also collected using scales. The valence, arousal, dominance, and liking scales are presented in Figure 6. The familiarity was collected on a Likert scale of 1-5, with 1 being "*very unfamiliar*" and 5 being "*very familiar*". Similarly, PANAS scale annotations for ten positive (Interested, Strong, Enthusiastic, Proud, Inspired, Determined, Alert, Attentive, Active) and ten negative (Distressed, Irritable, Guilty, Scared, Upset, Hostile, Jittery, Ashamed, Nervous, Afraid) emotions were also collected on a scale of 1-5, with 1 denoting "*very slightly or not at all*" to 5 denoting "*extremely*" for each emotion in the scale. To calculate the Positive Affect Score, we summed up the scores for positive items (Interested, Strong, Enthusiastic, Proud, Inspired, Determined, Alert, Attentive, and Active). This score can range from 10 to 50, with higher scores indicating higher levels of positive affect. For the Negative Affect Score, we have added the scores for negative items

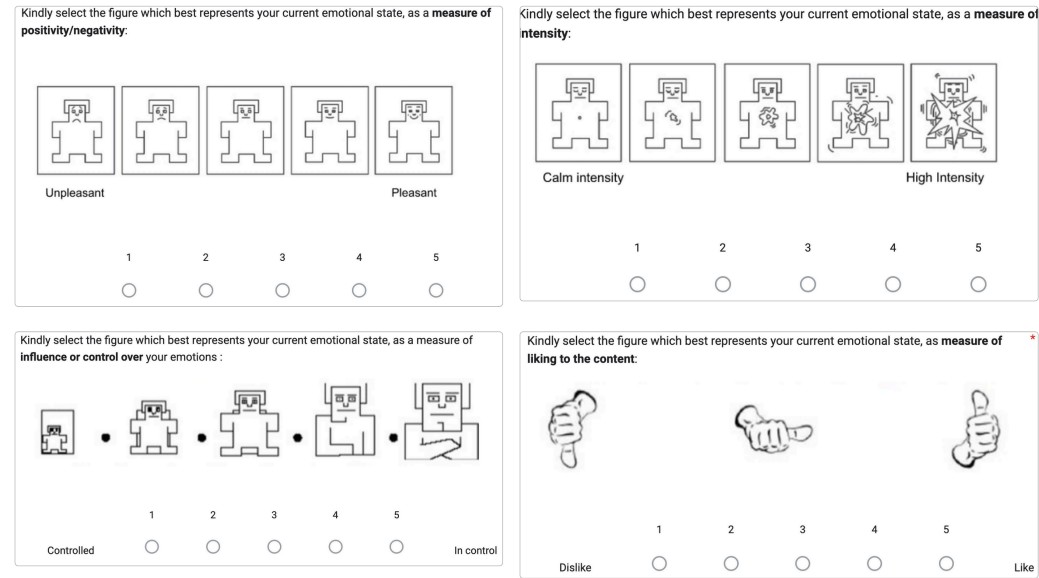

Figure 6: Illustration of self-assessment scales as following: Valence SAM, Arousal SAM, Dominance SAM, and Liking scale.

(Distressed, Irritable, Guilty, Scared, Upset, Hostile, Jittery, Ashamed, Nervous, Afraid). This score also ranges from 10 to 50, with lower scores indicating lower levels of negative affect.

### 4.4.1 GHQ-12

This study used a twelve-item General Health Questionnaire designed to measure non-psychotic mental health. This scale is rated on a 4-point scale with a timeframe of "in the last one week." We applied the Likert scoring method (0-1-2-3), where each of the four response options ("Not at all," "No more than usual," "Rather more than usual," "Much more than usual" or "Better than usual," "Same as usual," "Less than usual," "Much less than usual") is assigned a numerical value of 0, 1, 2, or 3. For each of the 12 questions, we summed the scores based on the responses given by the respondents. The total score can range from 0 to 36. A lower total score (closer to 0) indicates better mental health and lower psychological distress, while a higher total score (closer to 36) suggests higher levels of psychological distress and potential mental health issues.

### 4.4.2 Personality

The personality questionnaire, depicted in Figure 7 with item numbers, employs the BFI-10 (Big Five Inventory-10) scoring method to derive personality scores. This method involves assigning scores to each item based on the respondent's selection. For Extraversion, item 1 is reverse-scored (For reverse-scoring item, subtract the respondent's original score from the highest possible score on the scale plus one.), while item 5 is scored as is. In Agreeableness, item 2 is scored as is, and item 7 is reverse-scored. Conscientiousness is determined by reversing the score for item 3 and scoring item 8 as is. Neuroticism involves reversing the score for item 4 and scoring item 9 as is. Openness to Experience is evaluated by reversing the score for item 5 and scoring item 10 as is. By applying these scoring guidelines to each item, we calculate the total score for each trait.

### 4.4.3 VRSQ

The VRSQ questionnaire is illustrated in Figure 8 with question numbers. To determine the VRSQ score, we first calculated two sub-scores: A and B. Sub-score A is obtained by summing the responses to questions 1 through 4, while sub-score B is derived from questions 5 through 9. Then, to standardize these scores, A is divided by 12 and multiplied by 100 to yield C, and B is divided by 15 and multiplied by 100 to produce D. Finally, the VRSQ score is calculated as the average of C and D, providing a comprehensive measure of VR sickness for an individual Kim et al. (2018).

| I see myself as someone who … | Disagree strongly | Disagree a little | Neither agree nor disagree | Agree a little | Agree strongly |
|---|---|---|---|---|---|
| 1.  … is reserved | (1) | (2) | (3) | (4) | (5) |
| 2.  … is generally trusting | (1) | (2) | (3) | (4) | (5) |
| 3.  … tends to be lazy | (1) | (2) | (3) | (4) | (5) |
| 4.  … is relaxed, handles stress well | (1) | (2) | (3) | (4) | (5) |
| 5.  … has few artistic interests | (1) | (2) | (3) | (4) | (5) |
| 6.  … is outgoing, sociable | (1) | (2) | (3) | (4) | (5) |
| 7.  … tends to find fault with others | (1) | (2) | (3) | (4) | (5) |
| 8.  … does a thorough job | (1) | (2) | (3) | (4) | (5) |
| 9.  … gets nervous easily | (1) | (2) | (3) | (4) | (5) |
| 10. … has an active imagination | (1) | (2) | (3) | (4) | (5) |

Figure 7: Illustration of BFI-10 personality scale used for our experiment with item number.

| | | | | |
|---|---|---|---|---|
| 1. General discomfort | None | Slight | Moderate | Severe |
| 2. Fatigue | None | Slight | Moderate | Severe |
| 3. Headache | None | Slight | Moderate | Severe |
| 4. Eye strain | None | Slight | Moderate | Severe |
| 5. Difficulty focusing | None | Slight | Moderate | Severe |
| 6.  Fullness of the Head | None | Slight | Moderate | Severe |
| 7. Blurred vision | None | Slight | Moderate | Severe |
| 8. Dizziness with eyes closed | None | Slight | Moderate | Severe |
| 9. *Vertigo | None | Slight | Moderate | Severe |

Figure 8: Illustration of Virtual Reality Sickness scale with questions as used in our experiments.

# 5 Data Analysis and Experiments

## 5.1 Content Analysis

In Figure 9, we illustrate the frequency distribution of self-reported annotations for each scale. Our analysis showed that the self-reports are mostly unbalanced. For example, the valence label tends to skew towards positive values, while the arousal label is predominantly neutral. Additionally, most participants reported a high level of control over their emotions on the dominance scale. Most participants also indicated that they liked the content used to induce emotions, which may explain the low negative affect scores across the board. Familiarity with the content was mostly high among the participants. Positive affect scores were more evenly distributed than skewed negative affect scores. Additionally, we found the GHQ scores of participants are evenly distributed. We observed that due to the subjective nature of emotions, participants' high levels of liking and perceived control over their emotions likely contributed to their overall positive reports.

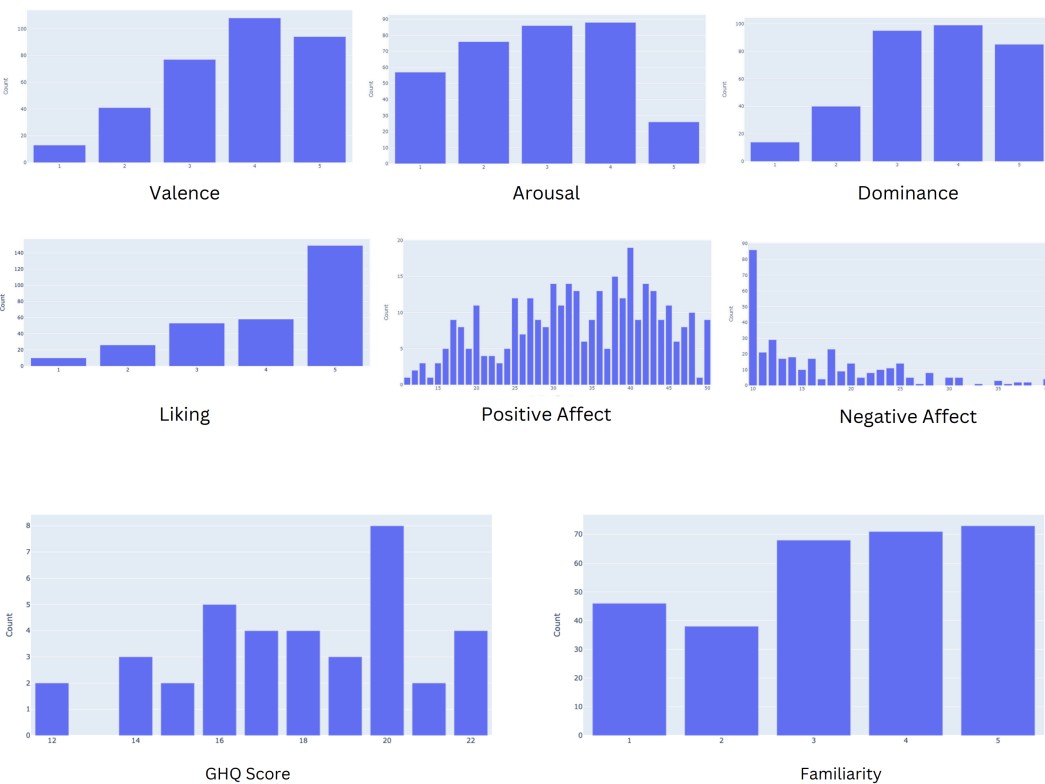

Figure 9: Frequency Distribution of self-reported annotations for Valence, Arousal, Dominance, Liking, Positive Affect, Negative Affect, GHQ Scores and Familiarity.

## 5.2 Data Cleaning

The physiological signal data was initially collected as ACQ files from Biopac, which allows the extraction of the data as text files. Before extraction, the signals were manually checked for errors, with any erroneous sections labeled for post-processing. The data was then downsampled using the software: EDA data to 15.625 Hz and PPG data to 125 Hz. The Biopac system was also used to calculate the BPM for the PPG data. After downsampling and BPM calculation, each participant's physiological signal text files and annotation files were downloaded. These files were then uploaded to Python using the pandas library and cleaned to remove all segments labeled as errors. The data was checked for NaN values and outliers, specifically PPG values outside the normal 35-140 BPM range and EDA values outside the 0-60 μS range. Following this filtering, the signal data was labeled with video ID, video name, playlist ID, and gender details based on timestamps. This helps us prepare the raw CSV files for further analysis.

## 5.3 Text Data Preparation

The textual descriptions were collected using a semi-structured interview technique, where an interviewer asked participants to explain their experiences qualitatively. Audio recordings were made for both the interviewer and interviewee. These recordings were then converted into text format using the Google Cloud Speech-to-Text API[5]. After conversion, the text data was manually checked for errors. Finally, the text data of participants' responses was extracted and compiled into a CSV file for further analysis.

## 5.4 Text Data Analysis

To assess the quality of our textual descriptions, we conducted a correlation analysis between these descriptions and the participant-reported valence and arousal (V/A) ratings (see Figure **??**). For this analysis, we first extracted text embeddings using DistilBERT and subsequently applied Principal Component Analysis (PCA) to reduce the dimensionality of these embeddings. The resulting principal components were then visualized using a heatmap to illustrate their relationship with the V/A labels. Our findings indicate a strong correlation between the first principal component (PC1) and the V/A ratings, suggesting that the textual data closely aligns with the self-reported labels.

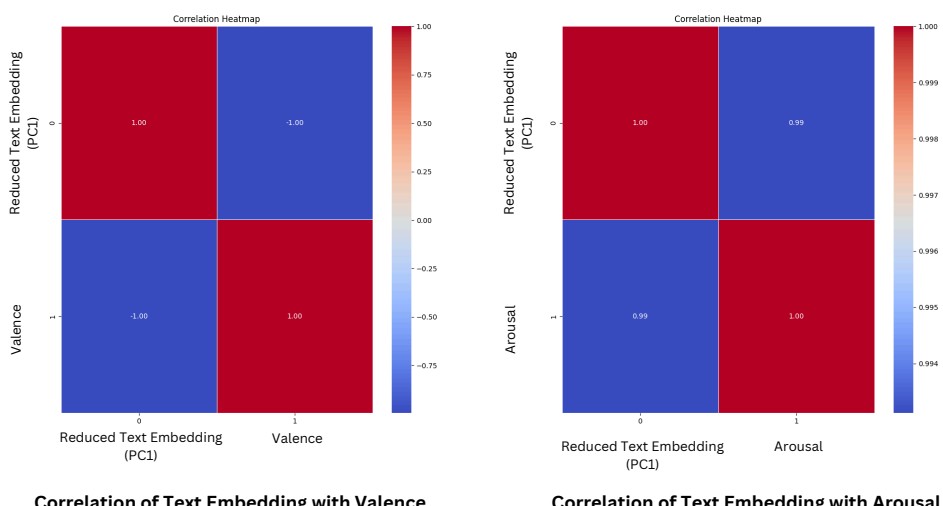

**Correlation of Text Embedding with Valence**   **Correlation of Text Embedding with Arousal**

Figure 10: Illustration of correlation between V/A labels and textual descriptors

## 5.5 Physiological Features

For EDA data following signal cleaning and signal decomposition into tonic and phasic components, we have manually extracted the time domain features, such as statistical features, SCR-specific, and frequency domain features, such as power band features, variance, range, skewness, kurtosis. Similarly, we have extracted features using the Neurokit library for PPG data following the filtering and winsorization. We extracted Heart Rate (HR), Heart Rate Variability (HRV) Time-Domain Features, and Heart Rate Variability (HRV) Frequency-Domain Features. Following feature extraction, we analyzed correlation and dropped features with high correlation. Table 4 mentions the final features selected for classification.

## 5.6 Experiment Details and Results Analysis

### 5.6.1 Experimental Setup

We have used a machine with an AMD EPYC 7763 64-core Processor CPU and NVIDIA A100 40GB GPU to train all our models. Training a classical machine learning model on physiological signals for

---
[5]https://cloud.google.com/speech-to-text

| Signal | Selected Features |
|--------|-------------------|
| PPG | 'BPM', 'IBI', 'PPG_Rate_Mean', 'HRV_MedianNN', 'HRV_Prc20NN', 'HRV_MinNN', 'HRV_HTI', 'HRV_TINN', 'HRV_LF', 'HRV_VHF', 'HRV_LFn', 'HRV_HFn', 'HRV_LnHF', 'HRV_SD1SD2', 'HRV_CVI', 'HRV_PSS', 'HRV_PAS', 'HRV_PI', 'HRV_C1d', 'HRV_C1a', 'HRV_DFA_alpha1', 'HRV_MFDFA_alpha1_Width', 'HRV_MFDFA_alpha1_Peak', 'HRV_MFDFA_alpha1_Mean', 'HRV_MFDFA_alpha1_Max', 'HRV_MFDFA_alpha1_Delta', 'HRV_MFDFA_alpha1_Asymmetry', 'HRV_ApEn', 'HRV_ShanEn', 'HRV_FuzzyEn', 'HRV_MSEn', 'HRV_CMSEn', 'HRV_RCMSEn', 'HRV_CD', 'HRV_HFD', 'HRV_KFD', 'HRV_LZC' |
| EDA | 'ku_eda', 'sk_eda', 'dynrange', 'slope', 'variance', 'entropy', 'insc', 'fd_mean', 'max_scr', 'min_scr', 'nSCR', 'meanAmpSCR', 'maxAmpSCR', 'meanRespSCR', 'sumAmpSCR', 'sumRespSCR' |

Table 4: List of Selected Features for Classification Tasks

any classification task (Arousal, Valence, and Stimulus-Label) took around 10-15 minutes of CPU time. Similarly, training the BERT-based text classification models took approximately 20 minutes of GPU time for each classification task. The Contrastive Language-Signal Pre-training (CLSP) Model required around 55.5 GPU hours for training 7 epochs in a Leave-One-Subject-Out (LOSO) setup for 37 participants. We trained the models for Electrodermal Activity (EDA) and Photoplethysmogram (PPG) signals separately, totaling 333 GPU hours for training 7 epochs across all tasks (Stimulus Label, Valence Label, and Arousal Label). Due to the CPU-intensive nature of our CLSP experiments and the limited CPU computing power available, our experiments took longer than expected.

### 5.6.2   Discussion on Physiological Baseline

Our baseline results for the Valence and Stimulus_Label classification task across all classical machine-learning models were better than random, suggesting that our models are able to separate features for these labels. We observed that stimulus labels are easy to predict using physiological signal-based features as compared to subjective labels like valence and arousal. The PPG+EDA features gave the best performance for valence classification. And EDA features provided the best performance for Stimulus_Label classification. The results were nearly random for Arousal classification even after performing duplicate upsampling, suggesting arousal classification is a difficult label to predict purely based on physiological signals-based features. We have visualized our EDA and PPG handcrafted features for all three tasks using t-SNE algorithms as shown in Figure 5.6.2. We observed that t-SNE features are separable for Stimulus_Label in the case of EDA data. While for other labels the features are overlapping. This suggests the need for more complex models and better representation learning for valence and arousal prediction.

### 5.6.3   Discussion on Text Baseline

To assess the quality of our text data, we performed all three classification tasks using only text data as our input. Text-based classification models significantly outperformed classification models trained on only physiological signals. This better performance is likely due to the use of large pre-trained embedding models. We used the DistilBERT and XLM-RoBERTa Base for classification, where DistilBERT performed better. We trained all models with a batch size of 16 over 7 epochs and a learning rate of 2e-5. Furthermore, we visualized the embeddings from our models and found that they are visually distinct, as illustrated in Figure 12.

### 5.6.4   Discussion on CLSP

To assess the significance of aligning textual descriptions to physiological signal data, we performed CLSP training for all three labels using EDA only, PPG only, and EDA+PPG handcrafted features (for 296 tasks excluding baseline) along with text data. To compare our CLSP results, we first trained a hand-crafted features-based neural network (HC+NN) model with two hidden linear layers of dimensions 50 and 100. These models were trained for a batch size of 32 with 200 epochs, using a learning rate of 0.001. For optimization, we utilized the Adam optimizer with beta values

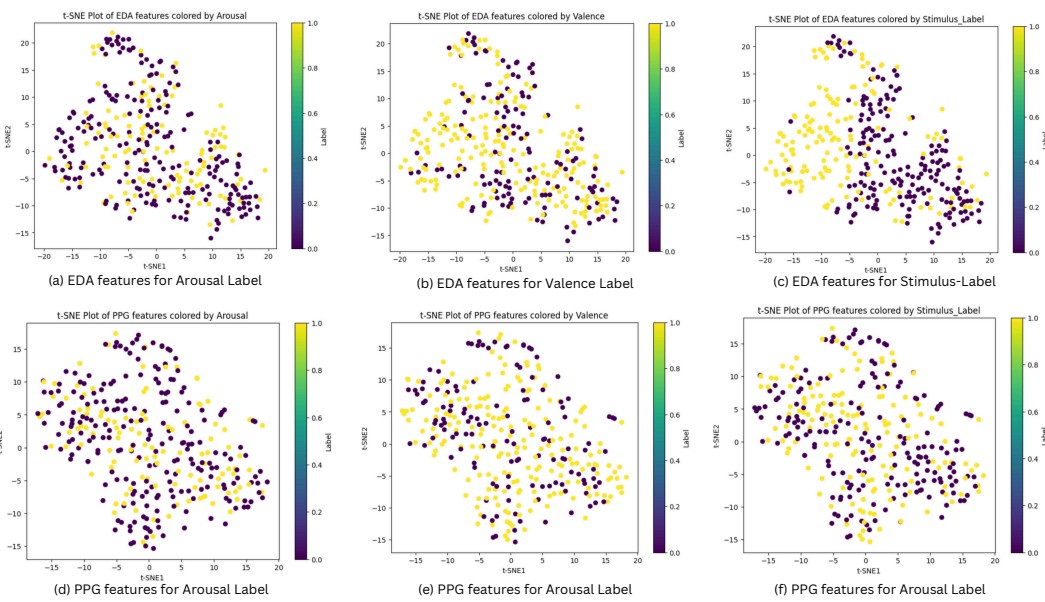

Figure 11: t-SNE plot depicting feature distribution of physiological signals according to various labels (Arousal, Valence, and Stimulus-Label).

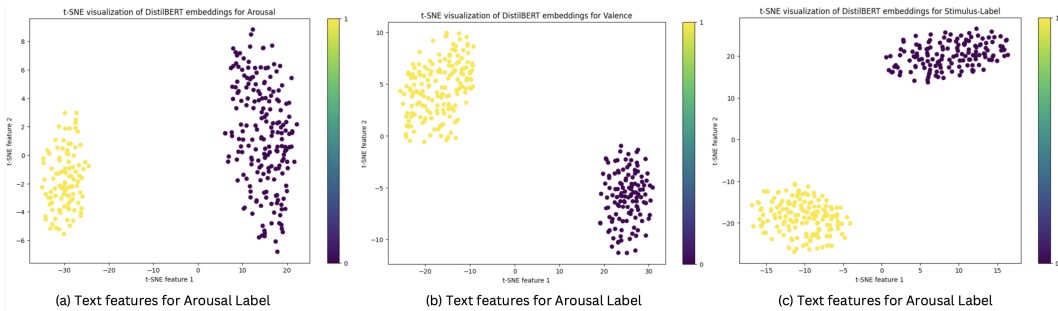

Figure 12: t-SNE plot depiction of Text data features for our three labels: Arousal, Valence, and Stimulus-Label

of 0.9 for beta1 and 0.999 for beta2 and an epsilon value of 1e-8. The CLSP model was then trained for contrastive objectives using the HC+NN model for physiological signal embeddings and the DistillBert model with a projection head of a single linear layer with dimension 100 for text embeddings. The project head was added to match the dimensionality of the text embeddings (originally 768) with signal embedding. The training was conducted for 15 epochs with a learning rate of 0.001 and batch size of 32. We found that our results were significantly better for arousal and valence labels, suggesting the importance of augmenting text data for training subjective labels like Arousal and Valence. The results were not as good for Stimulus_Labels, indicating that the features extracted from signals do not complement the features extracted from text. This mismatch could have happened due to the subjective nature of textual descriptions that Stimulus_Labels cannot capture. This misalignment might have led to ineffective contrastive training. The EDA+Text outperformed PPG+Text and PPG+EDA+Text, suggesting that EDA features might align more with textual descriptions for arousal and valence labels. We also observed that the early fusion of EDA and PPG features for CLSP has led to poor performance compared to EDA-only and PPG-only features, with Text indicating a need for designing a more complex fusion technique. Overall, our results suggest that aligning text data with physiological signals can improve learning for subjective labels, achieving results that cannot be attained with objective labels alone.