# OpenReview forum: "EEVR: A Dataset of Paired Physiological Signals and Textual Descriptions for Joint Emotion Representation Learning"
_NeurIPS.cc/2024/Datasets_and_Benchmarks_Track — NeurIPS 2024 Track Datasets and Benchmarks Poster_

### Official Review · Reviewer_UtWW · 2024-06-24
**EEVR: A Virtual Reality-Based Emotion Dataset Featuring Paired Physiological Signals and Textual Descriptions**

**Rating:** 9
**Confidence:** 4
**Clarity:** Yes, the paper is well-written.

**Review:**

The authors have presented a good justification of how the study is placed in the context of the larger literature. Primary among the originality of their work is the rigorous methods used in the experiment and mixed-methods data collection. The researchers understood that there needs to be a standardized procedure that takes into account previous VR exposure and makes sure that participants who are not familiar with VR are able to navigate the experiment. The qualitative interview at the end of the physiological data collection was a novel idea as it allowed to get better information on the emotions participants experienced. It is clear from the way in which the experiment was conducted that the researchers carefully thought out the process of baseline data collection.

**Strengths:**

The strengths of this research are as follows:
    1) Utilization of previously validated and reliable measures of emotion through the PANAS scale on 10 positive and 10 negative emotions.
    2) Inclusion of the SAM scale, again validated and reliable, to collect data on valence, arousal, and dominance.
    3) Sound experimental protocol with baseline data, familiarity with VR, stimulus exposure with breaks, and qualitative interview.
    4) Somewhat clear process of qualitative data transcribing.

**Additional Feedback:**

The only other feedback I have is with regards to the qualitative data collection process. While the researchers state that the data was transcribed and coded, how many coders were involved and is the raw qualitative data available? What were some of the themes that emerged when looking just at the qualitative data? Is that reported elsewhere?

**Correctness:**

I believe that the claims made in this submission are correct. The experimental design and methods were accurate and performed correctly.

**Documentation:**

Yes, there appears to be sufficient detail on the data collection process and organization not to mention ethical and responsible use of data from the study's beginning to end.

**Ethics:**

I do not suspect any ethical concerns.

**Limitations:**

The authors have addressed some of these limitations because not everyone adjusts well to virtual reality. This is important because while on one hand we are looking at involving VR into research, we must be cautious that it presents its own which are noted by the authors in the paper. One question to stimulate further thinking on this is how can we use the results from this study to improve mental health in the population. Self-report data is certainly biased and our emotions can fluctuate so how can we capture this using the information we have on hand?

**Opportunities For Improvement:**

One of the primary limitations of this work is the small sample size of only 37 participants. Similar studies may have more participants which only adds more to the dataset. Furthermore, it appears that the sample is limited to the 18-33 year-old age group and may not be representative of the larger population. Would the results hold if the sample was larger and the participant's age was more distributed? The results therefore are not generalizable.

**Relation To Prior Work:**

The authors have spent time discussing how this work differs from previous contributions so I have no further comment here.

**Summary And Contributions:**

This paper is looking at a virtual reality based emotion dataset in 37 participants using a mixed-methods research design. The paper highlights how emotion-related research sometimes relies on physiological methods of data collection alongside self-reports. However, there are a range of emotions that we experience on a day-to-day basis that are not fully captured. For example, we can use different tasks to measure stress but how do we measure acute stress or chronic stress? This study utilizes EEVR which uses 360 degree VR audiovisual stimuli as well as follow-up interview questions to gain a better understanding of emotional valence and arousal. This can further help enhance our understanding and way of approaching overall mental health status among individuals.

---

> ### Author Rebuttal · Authors · 2024-08-14
>
> Dear Reviewer,
>
> Thank you for providing encouraging feedback on our work. Below we have provided a response to your comments for further improvements:
>
> 1) **Sample Size and Generalization**: We acknowledge that our sample size and age range are small but it is comparable to participant size of other standard dataset, though we also aim to extend our dataset further by collecting more text-physiological signal pair data in the future for better generalisation and pre-training preferably in real-life settings.
>
> 2) **Implications for future research on mental health**: This is an interesting paradigm, and we aim to contribute significantly through our ongoing research. Currently, we are exploring several promising ideas, including leveraging everyday language and deviations in physiological signals from homeostasis (baseline physiology) to interpret and annotate emotional data. One focus is on incorporating layman-friendly textual descriptions of emotions. This could involve simpler and more intuitive forms of expression, such as abstract quotes, brief write-ups, or even emojis—methods that people commonly use in their daily lives for journaling or social interaction online. These simplified forms of expression can help in enhance data collection by making the process more human-centered, reducing intrusiveness, and integrating seamlessly into individuals' everyday routines. After our conversations with domain experts, we can see the utility of this approach in enabling passive monitoring of emotional fluctuations, particularly in individuals with minor mental health conditions like anxiety, depression, or PTSD. For these populations, tracking emotional states and their fluctuations is critical to both diagnosis and ongoing care, and the methods we are exploring could provide a low-effort, real-world solution for doing so without adding to the mental burden.
>
> 3) **Details about Qualitative Analysis**: Thank you for the additional feedback on our work. We are currently analyzing the qualitative data from our interviews. For our inductive thematic analysis process we have two coders who have independently open-coded the data, following axial and selective coding for our final themes. From our initial analysis we have identified few factors that could contribute to emotion elicitation in our participants. While we haven't released the raw data yet, we’re considering sharing the qualitative findings of our data collection in future iterations or publications. For now, the text descriptions in this work are raw responses from participants to our semi-structured interview questions which are used as additional data-source for better contextualization.

---

### Official Review · Reviewer_B6pf · 2024-07-24
**EEVR - Emotion recognition with physiological data and texttual descriptors**

**Rating:** 5
**Confidence:** 4
**Clarity:** kind of ok.

**Review:**

The dataset can help for the development of a novel understanding of ER with physiological signals.
Main limitations are related to the results of the textual descriptors, the VR validation, and data quality assessment.
In the following my questions:

In table 4, why different methods are used for the different combination of data? why not using all the data together (EDA+PPG+Text)? why the results with the text are worst? which kind of textual descriptors are used?

Another limitation is related to the proof that VR is well eliciting emotions.
Please, compare the same pipeline with other recent datasets (that use video or other methods, as Stroope Test, etc.) and compare the results, with the same binarization strategy.

Please, report the confusion matrix of the results to help in understanding the data quality. And show the correlation between V/A labels and textual descriptors. How much does the textual descriptors improve wrt the SAM?

**Strengths:**

The dataset can help for the development of a novel understanding of ER with physiological signals.

**Additional Feedback:**

please extend the signal characterization and results section.

**Correctness:**

the statement 'We show that augmenting our signals with self-reported textual annotations can improve performance on physiological signal-based emotion recognition tasks', of the abstract is not well supported by data

**Documentation:**

ok

**Ethics:**

ok

**Limitations:**

we are not sure that the emotions are well elicitated. There is the need to better show the data quality and validate the VR acquisition.

**Opportunities For Improvement:**

Try to use all the data together (EDA+PPG+Text)
Try to compare V/A and textual descriptors, with same methods.

**Relation To Prior Work:**

ok

**Summary And Contributions:**

The work is about data collection of physiological data, exploiting VR stimulation.

---

> ### Author Rebuttal · Authors · 2024-08-14
>
> Dear Reviewer,
>
> Thank you again for your thoughtful feedback. We have addressed your comments and believe these revisions will address your concerns and strengthen our work.
>
> 1) **Results and Methods Clarification**: We have reported results for all three combinations - EDA+Text, PPG+Text and  EDA+PPG+Text in Table 5 of supplementary (where the third row is EDA+PPG with no Text). Further, we have used the same underlying method for both with and without text baselines and physiological signals (as reported in Table 4 of the main paper and Table 5 of the supplementary). For the physiological signals-only baselines, we have used hand-crafted features + Linear layer, which is HC+NN, while for showing the importance of using paired textual descriptions, we have used the same HC+NN for extracting physiological signal embeddings and a pre-trained DistillBert model for text embeddings, after getting the embeddings for both physiological signals and text we have used contrastive training and thus we have called this method Contrastive Language-Signal Pre-training (CLSP) as explained in section 4.3 of main paper. We have shown results for all possible pairs for a clearer picture of differences in using physiological signals alone and combined with text. The results with text were comparable for both Arousal and Valence after running 7 epochs, while as shown in Table 5 supplementary on running for 15 epochs, we got significant improvement in results as compared to the without-text classifier (physiological signal only classifier) for both Arousal and Valence. While for Stimulus-label text descriptions didn’t improve the results due to the mismatch that could have happened due to the subjective nature of textual descriptions that Stimulus_Labels cannot capture, which we have also discussed in section 5.5.4 of supplementary. In our paper the textual descriptions are raw text data wherein participants have elaborated on their emotional experiences during VR exposure. More details on our textual descriptions are included in section 3.1 (5. Qualitative Interview) of the main paper and section 5.3 of the supplementary.
>
> 2) **Validation of VR Emotion Elicitation**: Thank you for recommending that we benchmark our dataset against established emotion elicitation methods, such as videos, psychological tests like the Stroop or TSST, and real-life stimuli. While conducting such benchmarking would require additional data collection from the same participants using multiple methods, we instead conducted further experiments to demonstrate the transferability of our pre-trained models on these datasets. We have performed zero-shot transferability of our models on three previously collected datasets - WESAD [6] (Elicitation using TSST Task)
> EMOGNITION [7] (Elicitation using Videos), and NURSE [8] (Real-life Stress setting within Hospital). The zero-shot transferability results (in the attached PDF) of our models to newer data domains without any fine-tuning suggests that the distribution of data collected from our VR tasks is comparable to data collected from prior methods, indicating that the VR tasks capture emotion data in a way that is consistent with traditional methods.
> Furthermore, we want to highlight the prior literature that supports the use of VR as an effective emotion-elicitation medium. Gilpin et al. [2] compared the physiological signature of sadness across text, film, and virtual reality, finding that participants in the VR experienced the highest degree of presence, with psychophysiological responses indicating an activating pattern of sadness (body's physiological response is more aroused), in contrast to the deactivating pattern observed in other mediums. Additionally, Smarathna et al. [3] reviewed the potential of VR to evoke emotions effectively and naturally, highlighting its ecological validity as a paradigm for studying emotions. Chirico et al. [4] also showed through the psychophysiological responses (BVP, SC, sEMG) that immersive video conditions would increase the intensity of awe experienced compared to 2D screen videos. Zimmer et al. also showed that TSST and TSST-VR have similar stress responses both in terms of self-reports and physiological responses. Similarly, Tian et al. [1] compared 2D versus 3D virtual reality environments and found that emotion stimulation was more intense in the 3D environment, as measured by EEG and skin conductance responses (SCR), due to the greater sense of presence. The VR literature on emotion elicitation [3] suggests VR as an effective medium for eliciting emotions within lab settings due to various factors like - high presence and high immersion that can help the participants in immersing well in the emotional stimuli, which might be difficult in the presence of experimenter in a normal lab-setting. Further, VR-based stimuli have the ability to simulate a wide variety of emotions that are more ecologically valid than psychological tests suggesting validity of our stimulus choice.
>
> 3) **Data Quality and correlation between V/A labels and textual descriptors**: We have presented the results of correlation graphs in the attached PDF. To show the correlation between text descriptions and V/A labels, we reduced the dimensionality of our text embedding (extracted using Distillbert) using PCA, and then we presented the heatmap. We found a high correlation between PC1 and V/A labels (as shown in pdf), suggesting that text data contributes positively to the classification task and improves the model's ability to predict labels (also visible in results table 5 of supplementary). The high correlation represents the quality of text descriptions in respect to the V/A labels.
>
> 4) We will revise the abstract to better reflect the findings of our study backed by data in the modified draft of our paper, wherein we will clarify more on how text-description paired physiological signals have performed compared with no-text-description-based binary classifications.

---

> > ### Author Response · Authors · 2024-08-14
> > **References**
> >
> > 1) Tian, F., Hua, M., Zhang, W., Li, Y., & Yang, X. (2021). Emotional arousal in 2D versus 3D virtual reality environments. PloS one, 16(9), e0256211.
> > 2) Gilpin G, Gain J, Lipinska G. The physiological signature of sadness: A comparison between text, film and virtual reality. Brain Cogn. 2021 Aug;152:105734. doi: 10.1016/j.bandc.2021.105734
> > 3) Somarathna, R., Bednarz, T., & Mohammadi, G. (2022). Virtual reality for emotion elicitation–a review. IEEE Transactions on Affective Computing.
> > 4) Chirico, A., Cipresso, P., Yaden, D.B. et al. Effectiveness of Immersive Videos in Inducing Awe: An Experimental Study. Sci Rep 7, 1218 (2017). https://doi.org/10.1038/s41598-017-01242-0
> > 5) Zimmer P, Wu CC, Domes G. Same same but different? Replicating the real surroundings in a virtual trier social stress test (TSST-VR) does not enhance presence or the psychophysiological stress response. Physiol Behav. 2019 Dec 1;212:112690. doi: 10.1016/j.physbeh.2019.112690.
> > 6) Schmidt, P., Reiss, A., Duerichen, R., Marberger, C., & Van Laerhoven, K. (2018, October). Introducing wesad, a multimodal dataset for wearable stress and affect detection. In Proceedings of the 20th ACM international conference on multimodal interaction (pp. 400-408).
> > 7) Saganowski, S., Komoszyńska, J., Behnke, M., Perz, B., Kunc, D., Klich, B., ... & Kazienko, P. (2022). Emognition dataset: emotion recognition with self-reports, facial expressions, and physiology using wearables. Scientific data, 9(1), 158.
> > 8) Hosseini, S., Gottumukkala, R., Katragadda, S., Bhupatiraju, R. T., Ashkar, Z., Borst, C. W., & Cochran, K. (2022). A multimodal sensor dataset for continuous stress detection of nurses in a hospital. Scientific Data, 9(1), 255.

---

> > ### Comment · Reviewer_B6pf · 2024-09-02
> >
> > Thank you for your reply. I keep my rating.
> > As a suggestion, highlight the best results in Table 2, combine Table 2 and 3, and report both accuracy and F1 in the same way (100% or 1). The results from Table 2 are in contradiction with Table 4, and too much close to a random prediction. The data distribution is missing, as well ad the confusion matrices, to see how much random is the model prediction and the data value.

---

> > ### Author Response · Authors · 2024-09-02
> >
> > Thank you for your reply. As per your suggestion, we will highlight the best results and combine Tables 2 and 3 in the final version of the paper. We kept them separate because Table 2 was for physiological-signal baselines, while Table 3 was for textual data baseline, and it suited the present structure of the paper.
> >
> > Regarding the results in Table 2 and Table 4, as mentioned in the rebuttal, we have shown improved results in the supplementary Table 5, where we got much improvement over physiological baselines and random results as we achieved the highest classification accuracy of 77.248% and F1 of 0.813  for arousal while 70.383% and F1 of 0.734  for valence which is much above random. Thus, it shows the improvement achieved by using textual descriptions along with physiological signals.
> >
> > We also wanted to clarify that within emotion recognition using physiological signals, each person has a unique physiological baseline and, thus, a separate data distribution.
> > So, the results presented are calculated using LOSO (Leave one subject out) validation techniques to ensure that models generalize well to new, unseen participants. Further, our model also generalised well for participants' data from previously available datasets collected in completely different settings. Suggesting the generalizability and usability of our dataset for training emotion classifiers.
> >
> > Further, we also present here the median of confusion matrix and the best confusion matrix for all 37 LOSO models in our case:
> >
> > -----------------EDA+TEXT-----------------------
> >
> > Input Data: EDA+Text  Class: Valence Result: Median Case
> > True Negatives (TN): 0 False Positives (FP): 2
> > False Negatives (FN): 0 True Positives (TP): 6
> >
> > Input Data: EDA+Text  Class: Valence Result: Best Case
> > True Negatives (TN): 8 False Positives (FP): 0
> > False Negatives (FN): 0 True Positives (TP): 0
> >
> > Input Data: EDA+Text  Class: Arousal Result: Median
> > True Negatives (TN): 0  False Positives (FP): 4
> > False Negatives (FN): 0 True Positives (TP): 4
> >
> > Input Data: EDA+Text  Class: Arousal Result: Best Case
> > True Negatives (TN): 8 False Positives (FP): 0
> > False Negatives (FN): 0  True Positives (TP): 0
> >
> > -----------------PPG+TEXT-----------------------
> >
> > Input Data: PPG+Text  Class: Valence Result: Median
> > True Negatives (TN): 3  False Positives (FP): 2
> > False Negatives (FN): 0 True Positives (TP): 3
> >
> > Input Data: PPG+Text  Class: Valence Result: Best Case
> > True Negatives (TN): 5 False Positives (FP): 0
> > False Negatives (FN): 0 True Positives (TP): 3
> >
> > Input Data: PPG+Text  Class: Arousal Result: Median
> > True Negatives (TN): 1 False Positives (FP): 2
> > False Negatives (FN): 0 True Positives (TP): 5
> >
> > Input Data:  PPG+Text  Class: Arousal Result: Best Case
> > True Negatives (TN): 7 False Positives (FP): 0
> > False Negatives (FN): 1 True Positives (TP): 0
> >
> > -----------------EDA+PPG+TEXT-----------------------
> >
> > Input Data:  EDA+PPG+Text  Class: Valence Result: Median
> > True Negatives (TN): 2  False Positives (FP): 1
> > False Negatives (FN): 2 True Positives (TP): 3
> >
> > Input Data:  EDA+PPG+Text  Class: Valence Result: Best Case
> > True Negatives (TN): 5 False Positives (FP): 0
> > False Negatives (FN): 1 True Positives (TP): 2
> >
> > Input Data:  EDA+PPG+Text  Class: Arousal Result: Median
> > True Negatives (TN): 4 False Positives (FP): 2
> > False Negatives (FN): 0 True Positives (TP): 2
> >
> > Input Data:  EDA+PPG+Text  Class: Arousal Result: Best Case
> > True Negatives (TN): 7 False Positives (FP): 0
> > False Negatives (FN): 1 True Positives (TP): 0

---

### Official Review · Reviewer_d4N1 · 2024-08-06
**Review of Submission 850**

**Rating:** 6
**Confidence:** 4

**Review:**

Pros:
* The paper is reasonably well-written with a clear structure, and appears to be of decent quality overall.
* The contributions, including the multi-modal dataset with physiological signals and aligned text descriptions, seem interesting as they are formulated in the paper, and there appear to be reasonable baselines for both sub-sets of the dataset (physiological, text) and the whole dataset (physiological and text).

Cons:
* The provided guidance and stated limitations, albeit somewhat useful, do not seem comprehensive enough nor particularly helpful with respect to the task of robust emotion recognition. Partly this is true because of an emphasis on controlled laboratory settings, which likely limit the generalization of the dataset and provided guidance to real-world applications that can include variables that affect emotional responses.
* Though text descriptions sound useful as contextualized in the paper at first, their subjective nature could easily introduce variability and bias as the ability of a participant's ability to articulate their emotions comes into play. Further analysis is needed in this regard, and with respect to the binary classification of emotions (e.g., are the evaluation datasets themselves limited or biased to self-reports in certain contexts?).
* The paper has no mentions of perception differences and their effects. Perception differences would surely would be impacted by the videos shown, the VR headset (in this case, Meta Quest Pro), and aspects of the procedure such as how the display may have been affected between participants (distracting debris, added moisture depending on the climate of the room, etc).

**Strengths:**

As mentioned above, the paper is reasonably well-written with a clear structure, and appears to be of decent quality overall. Furthermore, the contributions, including the multi-modal dataset itself, are interesting as formulated in the paper and include reasonable baselines.

**Additional Feedback:**

Some additional feedback:
* It would be interesting to include some literature, if available, and commentary, if applicable, about remote physiological sensing and whether or not the presence of such sensing versus contact or near-field approaches could cause changes in proposed emotion datasets and ultimately emotional responses by humans. I don't expect additional experiments or anything like that, more curious if the authors see a connection that could be interesting (based on existing literature).
* Datasets, and subsequent studies, motivated by virtual reality headsets seem to often ignore significant effects of perception differences in their analysis. I strongly encourage authors to dig deeper into literature (e.g., [1](https://www.frontiersin.org/journals/psychology/articles/10.3389/fpsyg.2015.00026/full), [2](https://journals.sagepub.com/doi/abs/10.1177/0963721411422522), [3](https://books.google.com/books?hl=en&lr=&id=5mfKEAAAQBAJ&oi=fnd&pg=PA134&dq=jeremy+bailenson+immersive+emotion&ots=BiTKEHjVhV&sig=LPNfCvE3ManM5gIl8dyDFWiD-rs#v=onepage&q=jeremy%20bailenson%20immersive%20emotion&f=false)) involving perception, emotion recognition, and virtual reality in order to summarize additional guidance on that matter, and to perhaps better motivate some of the listed limitations.
* Are the videos utilized (a list of them) provided somewhere for viewing and/or download? Not sure if I missed this somewhere in the paper.

**Clarity:**

The paper is reasonably well-written with a clear structure, and appears to be of decent quality overall.

**Correctness:**

The claims made are reasonable, albeit limited in some cases, and limitations are clearly noted in a separate section.

**Documentation:**

Yes, the corresponding details, in addition to various portions of the main paper, appear to be summarized in section 2.3 of the supplementary material.

**Ethics:**

As far as I can tell, there are no ethical concerns with the submission that warrant further discussion or review. This work involves human subjects, but it appears the authors have already followed existing protocols in their corresponding institution (i.e., IIT-Delhi IRB).

**Limitations:**

The authors listed numerous limitations in a meaningful manner, including VR-specific limitations such as sickness and discomfort, and noting participant bias. Perception, as noted in other parts of the review, is significantly affected by the use of virtual reality for a variety of reasons (as noted, comfort, and other factors that weren't explicitly noted such as display quality, field-of-view, Inter-Pupillary Distance (IPD) Adjustment, and user calibration). In turn, a user of a VR headset's display of emotions will likely be affected by their baseline perception differences, as well as their change in perception based on the VR headset. It is critical that the authors consider improving their work from this perspective, and at the very least incorporate insights from existing literature in order to show readers the impacts of perception differences and shifts in perception, on emotion.

**Opportunities For Improvement:**

Opportunities for improvement:
* The authors should provide further analysis with respect to differences in subjective text descriptions, and how they may compare to descriptions generated by SOTA emotion classification approaches, perhaps from an egocentric VR headset perspective or an exocentric camera perspective.
* Though not explicitly a con noted above, the lack of a visual modality and other modalities such as audio makes it difficult to infer further context about the environment in which users wore a headset and viewed videos, and despite the impact the environment would likely have on users.
* Perhaps by using [Diemer et al., 2015](https://www.frontiersin.org/journals/psychology/articles/10.3389/fpsyg.2015.00026/full) as a starting point, the authors should consider studying, evaluating, and noting perception differences that would be impacted by various factors and subsequently have a significant effect on the usefulness of this dataset.

**Relation To Prior Work:**

The authors appropriately discuss previous contributions, though as noted in the additional feedback there could be a more comprehensive overview of prior contributions, as well as the effectiveness of these prior contributions on emotion recognition, with respect to certain choices and limitations (e.g., contact versus remote physiological sensing). If there is a significant gap in the literature on that note, that would be good to note and provide brief commentary about as well.

**Summary And Contributions:**

In summary, the authors contributions are as follows:
* A multi-modal dataset in what appears to be a less constrained, immersive lab setting. The dataset includes aligned raw text descriptions of emotions felt, emotional self-reports, valence scores, and arousal scores.
* A detailed experimental procedure for capturing physiological responses and aligned text descriptions within the aforementioned lab setting.
* Guidelines for dataset usage, including baseline models to predict emotions, arousal, and valence using physiological signals + text

---

> ### Author Rebuttal · Authors · 2024-08-14
>
> Dear Reviewer,
>
> Thank you for your valuable feedback and for highlighting the strengths and areas for improvement in our work. We have addressed your concerns in our responses below:
>
> 1) **Guidance and Generalization to Real-World Applications**: We acknowledge the concern that the controlled laboratory setting might limit the generalizability of our dataset. However, we believe it can serve as a valuable foundation for two purposes: inspiring text labelling for in-the-wild data collection using layman language and providing a pre-training dataset for decision models to trigger self-assessment in real-life settings. Prior work in physiological signal-based emotion data collection often faces challenges in gathering emotion labels in the wild [8] and typically relies on pre-trained models or semi-supervised/self-supervised methods [1, 2].
> Our dataset, which includes contextualized textual descriptions of participants' emotions, enables baseline models trained with text supervision that could serve as decision model for real-life data collection. To demonstrate this, we have conducted additional experiments, showing our model's transferability across three previously available datasets. We have used the following three datasets to show the transferability of our pre-trained models - WESAD [6]  (Elicitation using TSST task) , EMOGNITION [7] (Elicitation using Videos), and NURSE [8] (Real-life Stress setting within Hospital). The results (as attached in PDF) suggest that the representations learned from our dataset are able to perform well on data collected in different settings including real-life setting [8] without additional fine-tuning, indicating that our model can serve as a good starting point for training on new data domains.
>
> 2) **Subjective Nature of Text Descriptions**: We agree that subjective text descriptions can introduce variability due to individual differences in emotional articulation. To overcome this, we have introduced the CLSP model in our paper, which is designed to handle variability in articulation and text length. CLSP tokenizes the text into a sequence of tokens, which are then passed through a transformer model (DistilBERT in our case) to generate a fixed-size embedding, regardless of text articulation. The transformer architecture is well-suited for handling sequences of varying lengths and different levels of articulation, capturing the nuances of both short, simple captions and longer, more detailed descriptions. The attention mechanism dynamically weighs the importance of different parts of the text. Additionally, the contrastive learning objective that aligns paired input embeddings encourages the model to focus on the core meaning of the text, making it robust to variations in expression. This approach, inspired by the pioneering work CLIP [3] in computer vision, has proven effective in dealing with subjective textual descriptions. Furthermore, we collected text descriptions through qualitative semi-structured interviews, guided by HCI research methods, conducted by an author trained in qualitative interviews. Participants were probed from different angles to articulate the presence and absence of emotions and the reasons behind them, thus minimizing articulation differences in our dataset.
>
> Further, in our binary emotion classification approach, text descriptions serve as an additional data source alongside physiological signals to prepare embeddings using contrastive training. During the evaluation, we use self-assessment labels like Stimulus-label, Valence, and Arousal, similar to prior work on SOTA emotion classification [2, 4]. Our approach has improved performance over using only physiological signals, as shown in Table 5 (wherein the third row is EDA+PPG) of the supplementary for both valence and arousal classification, suggesting the utility of adding text descriptions.
>
> 3) **Perception Differences and Effects on Dataset Usefulness**: We recognize the importance of perception differences in virtual reality contexts for emotion elicitation. To address these differences in our data collection, we have used the following measures: we selected participants from similar technology-friendly backgrounds and have included time for familiarity with VR for all participants. Further, we have collected self-reports on VR sickness, comfort, and ergonomics (using VRSQ and general discomfort questions) and data on personality traits, which are available in our dataset for more context. None of our participants reported high scores in VRSQ assessments. Further, we have also adjusted the field-of-view and Inter-Pupillary Distance (IPD) before each data collection session, and each experiment started after user confirmation. Additionally, our text descriptions include details on VR-specific perceptions like video quality, background audio, camera angles and video transitions, which also help provide contextual details during emotion classification. All our data collection occurred in a controlled lab environment with adjusted temperature and minimal noise to maintain consistency. Furthermore, our results (Table 5 of supplementary) suggested that the classification task is possible using our physiological signal data across all our participants, suggesting the usefulness of our data.
>
> 4) **Improved Literature Review**: Thank you for pointing it out; we will add more literature, as suggested, discussing various experiment choices and on the importance of perceptions differences and their impacts on emotions in the final version of the paper.
>
> 5) We did not include Visual and Audio modalities in our dataset because they are not privacy-preserving; however, we wanted to highlight that we have adopted similar lab settings (Lab room with maintained temperature and minimal noise) for all participants to maintain consistency.
>
> 6) We have included all videos in the VR_Application.zip file, as mentioned in Section 1.5 of the supplementary materials.

---

> > ### Author Response · Authors · 2024-08-14
> > **References**
> >
> > 1) Soowon Kang, Cheul Young Park, Auk Kim, Narae Cha, and Uichin Lee. 2022. Understanding Emotion Changes in Mobile Experience Sampling. In Proceedings of the 2022 CHI Conference on Human Factors in Computing Systems (CHI '22). Association for Computing Machinery, New York, NY, USA, Article 198, 1–14. https://doi.org/10.1145/3491102.3501944
> > 2) S. Saganowski, B. Perz, A. G. Polak and P. Kazienko, "Emotion Recognition for Everyday Life Using Physiological Signals From Wearables: A Systematic Literature Review," in IEEE Transactions on Affective Computing, vol. 14, no. 3, pp. 1876-1897, 1 July-Sept. 2023, doi: 10.1109/TAFFC.2022.3176135.
> > 3) Radford, A., Kim, J.W., Hallacy, C., Ramesh, A., Goh, G., Agarwal, S., Sastry, G., Askell, A., Mishkin, P., Clark, J. and Krueger, G., 2021, July. Learning transferable visual models from natural language supervision. In International conference on machine learning (pp. 8748-8763). PMLR.
> > 4) P. J. Bota, C. Wang, A. L. N. Fred and H. Plácido Da Silva, "A Review, Current Challenges, and Future Possibilities on Emotion Recognition Using Machine Learning and Physiological Signals," in IEEE Access, vol. 7, pp. 140990-141020, 2019, doi: 10.1109/ACCESS.2019.2944001.
> > 5) Teresa Hirzle, Maurice Cordts, Enrico Rukzio, Jan Gugenheimer, and Andreas Bulling. 2021. A Critical Assessment of the Use of SSQ as a Measure of General Discomfort in VR Head-Mounted Displays. In Proceedings of the 2021 CHI Conference on Human Factors in Computing Systems (CHI '21). Association for Computing Machinery, New York, NY, USA, Article 530, 1–14. https://doi.org/10.1145/3411764.3445361
> > 6) Schmidt, P., Reiss, A., Duerichen, R., Marberger, C., & Van Laerhoven, K. (2018, October). Introducing wesad, a multimodal dataset for wearable stress and affect detection. In Proceedings of the 20th ACM international conference on multimodal interaction (pp. 400-408).
> > 7) Saganowski, S., Komoszyńska, J., Behnke, M., Perz, B., Kunc, D., Klich, B., ... & Kazienko, P. (2022). Emognition dataset: emotion recognition with self-reports, facial expressions, and physiology using wearables. Scientific data, 9(1), 158.
> > 8) Hosseini, S., Gottumukkala, R., Katragadda, S., Bhupatiraju, R. T., Ashkar, Z., Borst, C. W., & Cochran, K. (2022). A multimodal sensor dataset for continuous stress detection of nurses in a hospital. Scientific Data, 9(1), 255.

---

> > ### Comment · Reviewer_d4N1 · 2024-08-17
> > **Response to Rebuttal by Authors**
> >
> > I appreciate the rebuttal and additional information from the authors. I've raised my rating (4->6) as a result.
> >
> > I still have some doubts, especially with regard to perception differences and how useful this dataset could be despite that, but I will admit it's impractical to ask the authors to really dig deep into some of these aspects given the limited amount of time available to them. I strongly suggest the authors take this into account nevertheless as they consider the overarching feedback toward their work, and perhaps consider studying perception differences and ultimately the effects of the VR headset more carefully as a part of future works.
> >
> > I should note also that not including visual and audio modalities as a part of the dataset in order to make it more privacy-preserving is a bit of a strange argument, especially since the focus of this paper doesn't appear to be toward privacy (barring some very brief mentions) and having such modalities available would be great for future works that try to show there are strong correlations between not only the physiological signals and the text descriptions, but also the corresponding visual modalities or audio modalities. Otherwise, it could be totally possible that the physiological signals and text descriptions provided are strongly correlated while similar signals and text descriptions produced in an attempt to perform data collection akin to the authors are not always strongly correlated.

---

> > > ### Author Response · Authors · 2024-08-18
> > > **Response to Reviewer d4N1 Comment**
> > >
> > > Thank you so much for your thoughtful feedback and for increasing the rating. We really appreciate it. We completely understand your concerns about perception differences and how they might impact the usefulness of the dataset. We agree that this is an area that deserves more attention. As you suggested, we plan to explore it further in future research, particularly how individual perception in VR can influence data collection.
> > > Regarding the addition of visual and audio modalities, we agree with your comment. Privacy was one of the initial reasons we held back from collecting these modalities. Additionally, capturing videos of facial expressions—commonly used in prior work—was challenging in our case, given that the stimulus exposure occurred in VR, which would have required a specialized setup. Further audio could have been an interesting modality as it added to physiological and textual descriptions. We’re definitely considering incorporating these modalities in the future. We believe that adding them could significantly enhance the dataset’s usefulness, especially for showing stronger multimodal correlations.

---

### Author Rebuttal · Authors · 2024-08-16

Dear Reviewers,

Thank you for your time and valuable feedback on our work. Below, we have summarized our rebuttal and addressed the common points raised by all reviewers:

1) **Generalization of our Dataset:**  Our dataset was collected in a lab setting using VR to provide more realistic experiences. In addition to physiological signal data, we focused on gathering detailed text descriptions where participants could freely elaborate on their emotional experiences, such as the presence or absence of emotions and the reasons behind them, using everyday language.
To validate the generalizability of our dataset, we conducted additional experiments to demonstrate that our baseline model has zero-shot transferability to new data domains, including those collected in real-life settings. For these experiments, we selected three datasets: WESAD (a well-known dataset collected using the TSST psychological task), Emognition (collected using 2D video stimuli), and NURSE (collected in real-life hospital settings).
As shown in the attached PDF, our pre-trained model was able to predict emotions in these new domains with accuracy above random chance. In some cases, our model even outperformed supervised baseline models. These results highlight the effectiveness of the text descriptions in learning representations that are transferable across various data domains, regardless of the setting, device used, or cultural background of the participants. We followed a standardized pipeline (including data cleaning, participant-wise normalization, feature extraction, and classification) to ensure fair comparisons in our experiments.

2) **Usability of our Dataset:** Our baseline results (Table 5 of supplementary (where the third row is EDA+PPG with no Text)) and transferability experiments suggest that our dataset contains physiological signal data that can reliably represent changing emotions and is transferable to data domains from previous methods. We hope these results help the reviewers recognize the usefulness of our dataset.  While we agree with the reviewers on the importance of shifting toward collecting more real-life data, as we mentioned in our response to Reviewer d4N1, we believe that baseline datasets like ours are crucial in easing the data collection process in real-life settings. One of the significant challenges, in addition to varying environments, is participants' willingness to report emotions accurately. Our use of layman textual descriptions can inspire future work in addressing this challenge.

We hope our responses satisfactorily address all the reviewers' concerns and look forward to positive results.

---

### Author Response · Authors · 2024-08-24

Dear Reviewers,

As the author-reviewer discussion period is nearing its deadline, we wanted to check if you have any remaining questions or concerns following our response. We would be happy to provide any additional information you may need. Additionally, we would greatly appreciate any suggestions for improvement you might have.

Thank you.

---

### Decision · Program_Chairs · 2024-09-26

**Decision:**

Accept (Poster)

**Comment:**

This paper presents a dataset of physiological measurement for virtual reality applications.  The dataset is interesting and novel.  The reviewers highlight that the number of subjects is small and this will limit the potential of the dataset due to individual differences; however,  on balance the dataset is a good contribution to the community and will have utility. I would suggest the authors consider updating Table 2/3 and think about whether 3 decimal place precision is really necessary.  The rebuttal addressed the reviewers comments satisfactorily.